# Scalable Neural Theorem Proving on Knowledge Bases and Natural Language

## Abstract

Reasoning over text and Knowledge Bases (KBs) is a major challenge for Artificial Intelligence, with applications in machine reading, dialogue, and question answering. Transducing text to logical forms which can be operated on is a brittle and error-prone process. Operating directly on text by jointly learning representations and transformations thereof by means of neural architectures that lack the ability to learn and exploit general rules can be very data-inefficient and not generalise correctly. These issues are addressed by Neural Theorem Provers (NTPs) (Rocktäschel & Riedel, 2017), neuro-symbolic systems based on a continuous relaxation of Prolog's backward chaining algorithm, where symbolic unification between atoms is replaced by a differentiable operator computing the similarity between their embedding representations. In this paper, we first propose Neighbourhood-approximated Neural Theorem Provers (NaNTPs) consisting of two extensions to NTPs, namely *a)* a method for drastically reducing the previously prohibitive time and space complexity during inference and learning, and *b)* an *attention mechanism* for improving the rule learning process, deeming them usable on real-world datasets. Then, we propose a novel approach for jointly reasoning over KB facts and textual mentions, by jointly embedding them in a shared embedding space. The proposed method is able to *extract rules* and *provide explanations*—involving both textual patterns and KB relations—from large KBs and text corpora. We show that NaNTPs perform on par with NTPs at a fraction of a cost, and can achieve competitive link prediction results on challenging large-scale datasets, including `WN18`, `WN18RR`, and `FB15k-237` (with and without textual mentions) while being able to provide explanations for each prediction and extract interpretable rules.

## 1 Introduction

The main focus in Artificial Intelligence is building systems that exhibit intelligent behaviour (Levesque, 2014). In particular, Natural Language Understanding (NLU) and Machine Reading (MR) aim at building models and systems with the ability to read text, extract meaningful knowledge, and actively reason with it (Etzioni et al., 2006; Hermann et al., 2015; Weston et al., 2015; McCallum et al., 2017a). This ability enables both the synthesis of new knowledge and the possibility to verify and update a given assertion. For example, given the following statement:

*The River Thames is in the United Kingdom.*

and the following supporting text:

*London is the capital and most populous city of England and the United Kingdom. Standing on the River Thames in the south east of the island of Great Britain, London has been a major settlement for two millennia.*

a reader can verify that the statement is consistent since *London is standing on the River Thames* and *London is in the United Kingdom*. Automated reasoning applied on text requires Natural Language Processing (NLP) tools capable of extracting meaningful knowledge from free-form text and compiling it into KBs (Niklaus et al., 2018). However, the compiled KBs tend to be incomplete, ambiguous, and noisy, impairing the application of standard deductive reasoners (Huang et al., 2005).

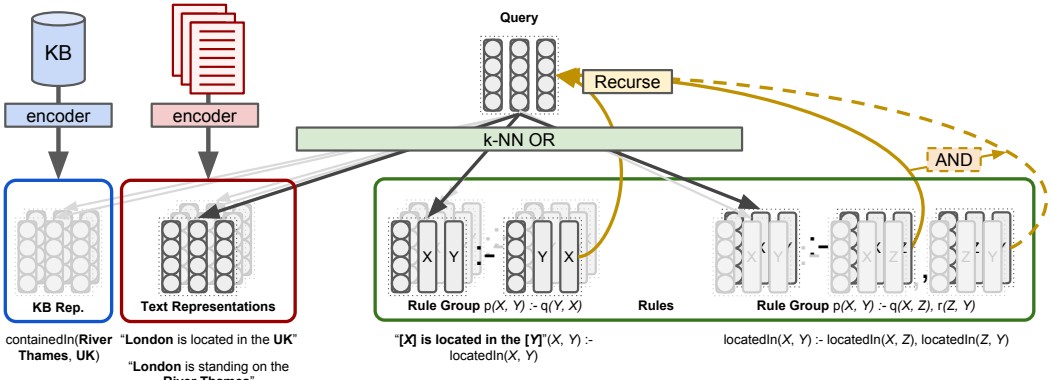

Figure 1: Overall architecture of NaNTPs: the two main contributions consist in a faster inference mechanism (represented by the κ-NN OR component, discussed in Section 3) and two dedicated encoders, one for KB facts and rules, and another for text (discussed in Section 4).

A rich and broad literature in MR has approached this problem within a variety of frameworks, including Natural Logic (MacCartney & Manning, 2007; Angeli & Manning, 2014) and Semantic Parsing (Dong & Lapata, 2016; Bos, 2008), and by framing the problem as Natural Language Inference—also referred to as Recognising Textual Entailment (Fyodorov et al., 2000; Condoravdi et al., 2003; Dagan et al., 2005; Bowman et al., 2015; Rocktäschel et al., 2015) —and Question Answering (Hermann et al., 2015). However, such methods suffer from several limitations. For instance, they rely on significant amounts of annotated data to suitably approximate the implicit distribution from which training and test data are drawn, and thus are often unable to generalise correctly in the absence of a sufficient quantity of training data or appropriate priors on model paramaters (e.g. via regularisation). Orthogonally, even if accurate, such methods also cannot provide explanations for a given prediction (Evans & Grefenstette, 2018; Marcus, 2018).

A promising strategy for overcoming these issues consists of combining *neural models* and *symbolic reasoning*, given their complementary strengths and weaknesses (Rocktäschel & Riedel, 2017; Evans & Grefenstette, 2018). While symbolic models can generalise well from a small number of examples when the problem domain fits the inductive biases presented by the symbolic system at hand, they are brittle and prone to failure when the observations are noisy and ambiguous, or when the domain's properties are not known or formalisable, all of which being the case for natural language (Raedt & Kersting, 2008). On the other hand, neural models are robust to noise and ambiguity but prone to overfitting (Marcus, 2018) and not easily interpretable (Lipton, 2018), making them incapable of providing explanations or incorporating background knowledge.

Recent work in neuro-symbolic systems (Garcez et al., 2015) has made progress in learning neural representations that allow for comparison of symbols not on the basis of identity, but of their semantics (as learned in continuous representations of said symbols), while maintaining interpretability and generalisation, thereby inheriting the best of both worlds. Among such systems, NTPs (Rocktäschel & Riedel, 2017) are end-to-end differentiable deductive reasoners based on Prolog's backward chaining algorithm, where unification between atoms is replaced by a differentiable operator computing their similarity between their embedding representations. NTPs are especially interesting since they allow learning *interpretable rules* from data, by back-propagating a KB reconstruction error to the rule representations. Furthermore, NTPs are *explainable*: by looking at the proof tree associated with the highest proof score, it is possible to know which rules are activated during the reasoning process—this enables providing explanations for a given reasoning outcome, performing error analysis, and driving modelling choices. So far, due to their computational complexity, NTPs have only been successfully applied to learning tasks involving very small KBs. However, most human knowledge is still stored in large KBs and natural language corpora, which are difficult to reason over automatically.

With this paper we aim at addressing these issues, by proposing:*a)* An efficient method for significantly reducing the time and space complexity required by NTPs by reducing the number of candidate proof scores and by using an attention mechanism for reducing the number of parameters required for learning new rules (Section 3), and *b)* An extension of NTPs towards text, by jointly embedding

predicates and textual surface patterns in a shared embedding space by means of an efficient reading component (Section 4).

## 2 Neural Theorem Proving

Mimicking backward chaining, NTPs recursively build a neural network enumerating all the possible proof states for proving a goal over the KB. NTPs rely on three modules for building this neural network; the Unification module, which compares subsymbolic representations of symbols, and mutually recursive OR and AND modules, which jointly enumerate all the proof paths, before the final aggregation choses the single, highest scoring state. We briefly overview these modules and the training procedure in the following.

**Unification Module.** In backward chaining, unification is the operator that matches two atoms like $\texttt{locatedIn}(\text{LONDON}, \text{UK})$ and $\texttt{situatedIn}(\text{X}, \text{Y})$. Discrete unification checks for equality between the elements of the atom (e.g. $\texttt{locatedIn} \neq \texttt{situatedIn}$) and binds variables to symbols via substitution (e.g. $\{\text{X}/\text{LONDON}, \text{Y}/\text{UK}\}$). Unification in NTPs matches two atoms by comparing their embedding representations via a differentiable similarity function, which enables matching different symbols with similar semantics, such as $\texttt{locatedIn}$ and $\texttt{situatedIn}$.

The $\texttt{unify}_{\boldsymbol{\theta}}(\text{H}, \text{G}, \text{S})$ operator creates a neural network module that does exactly that. Given two atoms $\text{H} = [\texttt{locatedIn}, \text{LONDON}, \text{UK}]$ and $\text{G} = [\texttt{situatedIn}, \text{X}, \text{Y}]$, and a proof state $S = (S_\psi, S_\rho)$ consisting of a set of substitutions $S_\psi$ and a proof score $S_\rho$, the $\texttt{unify}$ module compares the embedding representations $\boldsymbol{\theta}_{\texttt{locatedIn:}}$ and $\boldsymbol{\theta}_{\texttt{situatedIn:}}$ with a Radial Basis Function (RBF) kernel $k$, updates the variable binding substitution set $S'_\psi = S_\psi \cup \{\text{X}/\text{LONDON}, \text{Y}/\text{UK}\}$, and calculates the new proof score $S'_\rho = \min\left(S_\rho, k\left(\boldsymbol{\theta}_{\texttt{locatedIn:}}, \boldsymbol{\theta}_{\texttt{situatedIn:}}\right)\right)$. The resulting proof state $S' = (S'_\psi, S'_\rho)$ is further expanded with the $\texttt{or}$ and $\texttt{and}$ modules.

**OR Module.** The $\texttt{or}$ module unifies the goal with all the facts and rules in a KB. Concretely, for each rule H :– B in KB $\mathfrak{K}$, $\texttt{or}_{\boldsymbol{\theta}}^{\mathfrak{K}}(\text{G}, d, S)$ unifies the goal G with the rule head H, and invokes the $\texttt{and}$ module to prove atoms in the body B of the rule H, keeping track of the maximum proof depth $d$. For example, given a goal $\text{G} = [\texttt{situatedIn}, \text{Q}, \text{UK}]$, a rule H :– B with $\text{H} = [\texttt{locatedIn}, \text{X}, \text{Y}]$ and $\text{B} = [[\texttt{locatedIn}, \text{X}, \text{Z}], [\texttt{locatedIn}, \text{Z}, \text{Y}]]$, the model would unify the goal G with the head H of the rule, and instantiate $\texttt{and}$ modules, to prove sub-goals in the body G of the rule. Note that each fact F can be represented as a rule F :– [] with no body atoms.

**AND Module.** The $\texttt{and}$ module recursively tries to prove a list of sub-goals for a rule body. Concretely, given the first sub-goal G and the following sub-goals $\mathbb{G}$, the $\texttt{and}_{\boldsymbol{\theta}}^{\mathfrak{K}}(\text{G} : \mathbb{G}, d, S)$ module will substitute variables in G with constants according to the substitutions in S, and invoke the $\texttt{or}$ module on G. The resulting state be used to prove the atoms in $\mathbb{G}$, by recursively invoking the $\texttt{and}$ module. For example, when invoked on the rule body B mentioned above, the $\texttt{and}$ module will first substitute variables with constants for the sub-goal $[\texttt{locatedIn}, \text{X}, \text{Z}]$ and invoke the $\texttt{or}$ module, whose resulting state will be the basis of the next invocation of $\texttt{and}$ module on $[\texttt{locatedIn}, \text{Z}, \text{Y}]$.

**Proof Aggregation.** After building a neural network that enumerates all the proof paths of the goal G on a KB $\mathfrak{K}$, NTPs select the proof path with the maximum proof score:

$$\texttt{ntp}_{\boldsymbol{\theta}}^{\mathfrak{K}}(\text{G}, d) = \underset{\substack{S \,\in\, \texttt{or}_{\boldsymbol{\theta}}^{\mathfrak{K}}(\text{G}, d, (\varnothing, 1)) \\ S \neq \texttt{FAIL}}}{\arg\max} \; S_\rho$$

where $d$ is a predefined maximum proof depth. The initial proof state is set to $(\varnothing, 1)$, an empty substitution set, and a proof score of 1.

**Training** In NTPs, predicate and constant embeddings are learned by optimising a cross-entropy loss on the final proof score, by iteratively masking facts in the KB and trying to prove them using available facts and rules (Rocktäschel & Riedel, 2017). Negative examples are sampled from the positive ones by corrupting the entities (Nickel et al., 2016).

Other than learning embeddings of predicates and constants, NTPs can also learn *interpretable rules* from data. Rocktäschel & Riedel (2017) show that it is possible to learn rules from data by

specifying *rule templates* H :– B, with H $= [\boldsymbol{\theta}_{p:}, X, Y]$ and B $= [[\boldsymbol{\theta}_{q:}, X, Z], [\boldsymbol{\theta}_{r:}, Z, Y]]$, where $\boldsymbol{\theta}_{p:}, \boldsymbol{\theta}_{q:}, \boldsymbol{\theta}_{r:} \in \mathbb{R}^k$ are free parameters. Note that $\boldsymbol{\theta}_{p:}, \boldsymbol{\theta}_{q:}, \boldsymbol{\theta}_{r:}$ can be *learned from data*, and decoded by searching the closest representation of known predicates.

## 3   NEURAL THEOREM PROVING AT SCALE

**Scaling up Inference.**   The model in Section 2 is capable of deductive reasoning, and the proof paths with the highest score can provide human-readable explanations for a given prediction. However, a significant computational bottleneck lies in the `or` operator.

For instance, assume a KB $\mathfrak{K}$, composed of $|\mathfrak{K}|$ facts and no rules. The number of facts in a real-world KB can be quite large—for instance, Freebase contains 637 million facts (Dong et al., 2014), while the Google Knowledge Graph contains 18 billion facts (Nickel et al., 2016). Given a query G, in the absence of rules, NTP reduces to solving the following optimisation problem:

$$\text{ntp}_{\boldsymbol{\theta}}^{\mathfrak{K}}(G, 1) = \underset{F \in \mathfrak{K}, \ S \neq \texttt{FAIL}}{\arg\max} \ S_\rho \ \text{with } S = \texttt{unify}_{\boldsymbol{\theta}}(F, G, (\varnothing, 1)) \tag{1}$$

that is, it finds the fact $F \in \mathfrak{K}$ in the KB $\mathfrak{K}$ that, unified with the goal G, yields the maximum unification score. Recall from Section 2 that the unification score between a fact $F = [F_p, F_s, F_o]$ and a goal $G = [G_p, G_s, G_o]$ is given by the similarity of their representations in a Euclidean space:

$$\texttt{unify}_{\boldsymbol{\theta}}(G, F, (\varnothing, 1)) = \min\left\{1, k(\boldsymbol{\theta}_{G_p:}, \boldsymbol{\theta}_{F_p:}), k(\boldsymbol{\theta}_{G_s:}, \boldsymbol{\theta}_{F_s:}), k(\boldsymbol{\theta}_{G_o:}, \boldsymbol{\theta}_{F_o:})\right\} \tag{2}$$

where $k$ denotes a RBF kernel, and $\boldsymbol{\theta}_{G_p:}, \boldsymbol{\theta}_{G_s:}, \boldsymbol{\theta}_{G_o:} \in \mathbb{R}^k$ (resp. $\boldsymbol{\theta}_{F_p:}, \boldsymbol{\theta}_{F_s:}, \boldsymbol{\theta}_{F_o:} \in \mathbb{R}^k$) denote the embedding representation of the predicate, first and second argument of the goal G (resp. fact F).

Given a goal G, the NTPs proposed by (Rocktäschel & Riedel, 2017) will compute the unification score in Eq. 2 between G and every fact $F \in \mathfrak{K}$ in the KB. This is problematic, since computing the similarity between the representations of the goal G and every fact $F \in \mathfrak{K}$ is computationally prohibitive—the number of comparisons is $\mathcal{O}(|\mathfrak{K}|n)$, where $n$ is the number of goals and sub-goals in the proving process. However, $\text{ntp}_{\boldsymbol{\theta}}^{\mathfrak{K}}(G, d)$ only returns the single largest proof score, implying that every lower scoring proof is discarded during both inference and training.

One of the core contributions in this paper is to exactly compute $\text{ntp}_{\boldsymbol{\theta}}^{\mathfrak{K}}(G, m)$ by only considering a subset of proof scores that contains the largest one. Specifically, we make the following observation: given a goal G, if we know the most similar fact $F \in \mathfrak{K}$ in embedding space as measured by $\texttt{unify}_{\boldsymbol{\theta}}$, the number of comparisons needed for computing the final proof score $\text{ntp}_{\boldsymbol{\theta}}^{\mathfrak{K}}(G, 1)$ is reduced from $\mathcal{O}(|\mathfrak{K}|)$ to $\mathcal{O}(1)$. The same reasoning can be extended to rules as well.

We argue that, given G, we can restrict the search of the closest fact $F \in \mathfrak{K}$ to a Euclidean *local neighbourhood* of size $n$ of G, $\mathcal{N}_{\mathfrak{K}}(G) \subseteq \mathfrak{K}$ such that $|\mathcal{N}_{\mathfrak{K}}(G)| = n$, defined as follows: [1]

$$\mathcal{N}_{\mathfrak{K}}(G) = \text{k-}\underset{F \in \mathfrak{K}}{\arg\min} \|\boldsymbol{\theta}_{F:} - \boldsymbol{\theta}_{G:}\|_2 \tag{3}$$

Then, the matching fact F will be very likely to be contained across the $n$ most similar facts:

$$\text{ntp}_{\boldsymbol{\theta}}^{\mathfrak{K}}(G, 1) \approx \underset{\substack{S = \texttt{unify}_{\boldsymbol{\theta}}(F, G, (\varnothing, 1)) \\ F \in \mathcal{N}_{\mathfrak{K}}(G), \ S \neq \texttt{FAIL}}}{\arg\max} \ S_\rho \tag{4}$$

where the size of the neighbourhood is much lower than the size of the whole KB, *i.e.*, $|\mathcal{N}_{\mathfrak{K}}(G)| \ll |\mathfrak{K}|$. The same idea can be extended from facts to rules, by selecting only the rules H :– B $\in \mathfrak{K}$ such that their head H is closer to the goal. However, finding the *exact* neighbourhood of a point in a Euclidean space is very costly, due to the *curse of dimensionality* (Indyk & Motwani, 1998). Experiments showed that methods for identifying the exact neighbourhood can rarely outperform brute-force linear scan methods when dimensionality is high (Weber et al., 1998).

A practical solution consists in Approximate Nearest Neighbour Search (ANNS) algorithms, which focus on finding an *approximate* solution to the $k$-nearest neighbour search problem outlined in Eq. 3

---

[1] We approximate the neighbourhood with respect to the minimum of component distances with the neighbourhood with respect to a distance of concatenated representations.

on high dimensional data. Several families of ANNS algorithms exist, such as Locally-Sensitive Hashing (Andoni et al., 2015), Product Quantisation (Jégou et al., 2011; Johnson et al., 2017), and Proximity Graphs (Malkov et al., 2014).

In this work, we use Hierarchical Navigable Small World (HNSW) (Malkov & Yashunin, 2016), a graph-based incremental ANNS structure which can offer significantly better logarithmic complexity scaling during neighbourhood search than other approaches (Li et al., 2016). Specifically, given a subset of the KB $\mathcal{P} \subseteq \mathfrak{K}$—for instance, containing all facts in $\mathfrak{K}$—we construct a HNSW graph for all elements in $\mathcal{P}$, which has a $\mathcal{O}(|\mathcal{P}| \log |\mathcal{P}|)$ time complexity. Then, given a goal G, the HNSW graph is used for identifying its neighbourhood $\mathcal{N}_{\mathcal{P}}(G)$ within $\mathcal{P}$, which has a $\mathcal{O}(\log |\mathcal{P}|)$ time complexity. In our implementation, we construct the HNSW graph-based indexing structure when instantiating the model and, during training, we update the index every $b$ batches.

Specifically, in our implementation, we generate a partitioning $\mathfrak{P} \in 2^{\mathfrak{K}}$ of the KB $\mathfrak{K}$, where each element in $\mathfrak{P}$ groups all facts and rules in $\mathfrak{K}$ sharing the same signature. Then, we redefine the `or` operator as follows:

$$\mathtt{or}_{\boldsymbol{\theta}}^{\mathfrak{K}}(G, d, S) = [S' \mid S' \in \mathtt{and}_{\boldsymbol{\theta}}^{\mathfrak{K}}(B, d, \mathtt{unify}_{\boldsymbol{\theta}}(H, G, S)), H :\!- B \in \mathcal{N}_{\mathcal{P}}(G), \mathcal{P} \in \mathfrak{P}] \quad (5)$$

where, instead of trying to unify a goal or sub-goal G with all facts and rule heads in the KB, we constrain the unification with ANNS to only facts and rule heads in its local neighbourhood $\mathcal{N}_{\mathfrak{K}}(G)$.

**Improving Rule Learning via Attention.** Although NTPs can be used for *learning interpretable rules* from data, the solution proposed by Rocktäschel & Riedel (2017) can be quite data-inefficient, as the number of parameters associated to a rule can be quite large. For instance, assume the rule H :− B, with H = $[\boldsymbol{\theta}_{p:}, X, Y]$ and B = $[[\boldsymbol{\theta}_{q:}, X, Z], [\boldsymbol{\theta}_{r:}, Z, Y]]$ discussed in Section 2, where $\boldsymbol{\theta}_{p:}, \boldsymbol{\theta}_{q:}, \boldsymbol{\theta}_{r:} \in \mathbb{R}^k$. Such a rule introduces $3k$ parameters in the model, and it may be computationally inefficient to learn each of the embedding vectors.

We propose using an *attention mechanism* (Bahdanau et al., 2015) for attending over known predicates for defining the predicate embeddings $\boldsymbol{\theta}_{p:}, \boldsymbol{\theta}_{q:}, \boldsymbol{\theta}_{r:}$. Let $\mathcal{R}$ be the set of known predicates, and let $R \in \mathbb{R}^{|\mathcal{R}| \times k}$ be a matrix representing the embeddings for the predicates in $\mathcal{R}$. We define the $\boldsymbol{\theta}_{p:}$ as:

$$\boldsymbol{\theta}_{p:} = \mathrm{softmax}(\mathbf{a}_{p:})^{\mathsf{T}} R \quad (6)$$

where $\mathbf{a}_{p:} \in \mathbb{R}^{|\mathcal{R}|}$ is a set of trainable *attention weights* associated with the predicate $p$. This sensibly improves the parameter efficiency of the model in cases where the number of known predicates is low, *i.e.* $|\mathcal{R}| \ll k$, by introducing $c|\mathcal{R}|$ parameters for each rule rather than $ck$, where $c$ is the number of trainable predicate embeddings in the rule.

## 4 JOINTLY REASONING ON KNOWLEDGE BASES AND TEXT

In this section, we show we can use NaNTPs for jointly reasoning over KBs and natural language corpora. In the following, we assume that our KB $\mathfrak{K}$ is composed by facts, rules, and *mentions*. A fact is composed by a predicate symbol and a sequence of arguments, *e.g.* [locationOf, LONDON, UK]. On the other hand, a *mention* is a textual pattern between two co-occurring entities in the KB (Gabrilovich et al., 2013; Toutanova et al., 2015), such as "LONDON is located in the UK".

We represent mentions jointly with facts and rules in $\mathfrak{K}$ by considering each textual surface pattern linking two entities as a new predicate, and embedding it in a $d$-dimensional space by means of an end-to-end differentiable reading component. For instance, the sentence "United Kingdom borders with Ireland" is translated into the following mention in $\mathfrak{K}$: [[[arg1], borders, with, [arg2]], UK, IRELAND] by first identifying sentences or paragraphs containing KB entities, and then considering the textual surface pattern connecting such entities as an extra relation type.

While predicates in $\mathcal{R}$ are encoded by a look-up operation to a predicate embedding matrix $R \in \mathbb{R}^{|\mathcal{R}| \times k}$, textual surface patterns are encoded by an `encode`$_{\boldsymbol{\theta}}$ module. The signature of `encode`$_{\boldsymbol{\theta}}$ is $\mathcal{V}^* \to \mathbb{R}^k$, where $\mathcal{V}$ is the vocabulary of words and symbols occurring in textual surface patterns: it takes a sequence of tokens, and maps it to a $k$-dimensional embedding space.

More formally, given a textual surface pattern $t \in \mathcal{V}^*$—for instance, $t =$ [[arg1], borders, with, [arg2]]—the `encode`$_{\boldsymbol{\theta}}$ module first encodes each token $w$

in $t$ by means of a token embedding matrix $V \in \mathbb{R}^{|\mathcal{V}| \times k'}$, resulting in a pattern matrix $W_t \in \mathbb{R}^{|t| \times k'}$. Then, the module produces a textual surface pattern embedding vector $\boldsymbol{\theta}_{t:} \in \mathbb{R}^k$ from $W_t$ by means of an end-to-end differentiable encoder. In this paper, we use a simple $\texttt{encode}_{\boldsymbol{\theta}}$ module that computes the average of the token embedding vectors composing a textual surface pattern:

$$\texttt{encode}_{\boldsymbol{\theta}}(t \in \mathcal{V}^*) = \frac{1}{|t|} \sum_{w \in t} V_{w\cdot} \in \mathbb{R}^k$$

Albeit the encoder $\texttt{encode}$ can be implemented by using other differentiable architectures, such as Recurrent Neural Networks (RNNs), we opted for a simple averaging model, for the sake of simplicity and efficiency, knowing that such a model performs on par or better than more complex models, thanks to a lower tendency to overfit to training data (White et al., 2015; Arora et al., 2017).

## 5  RELATED WORK

A significant corpus of literature aims at addressing the limitations of neural architectures in terms of generalisation and reasoning abilities. A line of research consists of enriching neural network architectures with a differentiable *external memory* (Sukhbaatar et al., 2015; Graves et al., 2014; Joulin & Mikolov, 2015; Grefenstette et al., 2015; Kaiser & Sutskever, 2016; Miller et al., 2016; Graves et al., 2016). The underlying idea is that a neural network can learn to represent and manipulate complex data structures, thus disentangling the algorithmic part of the process from the representation of the inputs.

Another way of improving the generalisation and extrapolation abilities of neural networks consists of designing architectures capable of learning general, reusable *programs*—atomic primitives that can be reused across a variety of environments and tasks (Reed & de Freitas, 2016; Neelakantan et al., 2016; Parisotto et al., 2016). By doing so, it becomes also possible to train such models from enriched supervision signals, such as from *program traces* rather than simple input-output pairs.

Yet another line of work is *differentiable interpreters*—program interpreters where declarative or procedural knowledge, *e.g.*, a sorting program, is compiled into a neural network architecture (Bošnjak et al., 2017; Gaunt et al., 2016; Rocktäschel & Riedel, 2017; Evans & Grefenstette, 2018)—NTPs fall in this category. This family of models allows imposing strong inductive biases on the models by partially defining the program structure used for constructing the network, *e.g.*, in terms of instruction sets or rules. A major problem with differentiable interpreters, however, is their computational complexity, that so far deemed them unusable except for smaller-scale learning problems.

This work is also related to Rae et al. (2016), which use an approximate nearest neighbour data structure for sparsifying read operations in memory networks. Furthermore, Riedel et al. (2013) pioneered the idea of jointly embedding KB facts and textual mentions in a shared embedding space, by considering mentions as additional relations in a KB factorisation setting. This idea was later extended to more elaborate mention encoders by McCallum et al. (2017b). Our work is also related to path encoding models (Das et al., 2016) and random walk approaches (Lao et al., 2011; Gardner et al., 2014), which both lack rule induction mechanisms. Lastly, our work is related to Yang et al. (2017) which is a scalable rule induction approach for knowledge base completion, but has not been applied to textual surface patterns.

## 6  EXPERIMENTS

### 6.1  DATASETS AND EVALUATION PROTOCOLS

We report the results of experiments on benchmark datasets — Countries (Bouchard et al., 2015), Nations, UMLS, and Kinship (Kemp et al., 2006) — following the same evaluation protocols as Rocktäschel & Riedel (2017). Furthermore, since our scalability improvements described in Section 3 allow us to experiment on significantly larger datasets, we also report results on the WN18 (Bordes et al., 2013), WN18RR (Dettmers et al., 2018) and FB15k-237 (Toutanova et al., 2015) datasets— whose characteristics are outlined in Table 5. For evaluating our natural language reading component proposed in Section 4, we use FB15k-237.E (Toutanova et al., 2015)—the FB15k-237 dataset augmented with a set of textual mentions for all entity pairs derived from ClueWeb12 with Freebase entity mention annotations Gabrilovich et al. (2013)—and a set of manually generated mentions

Table 1: Link prediction results for WN18, WN18RR, and FB15k-237.E. Results for DistMult, ComplEx, ConvE, NeuralLP, and MINERVA are from Dettmers et al. (2018); Das et al. (2017). For the sake of comparison, we also trained ComplEx and DistMult with embedding size $d = 100$ and for 100 epochs, as NaNTP.

| | **WN18** | | | | **WN18RR** | | | | **FB15k-237.E** | | | |
| | | Hits | | | | Hits | | | | Hits | | |
| | MRR | @10 | @3 | @1 | MRR | @10 | @3 | @1 | MRR | @10 | @3 | @1 |
|---|---|---|---|---|---|---|---|---|---|---|---|---|
| DistMult (Yang et al., 2015) | 0.822 | 0.936 | 0.914 | 0.728 | 0.430 | 0.490 | 0.440 | 0.390 | 0.241 | 0.419 | 0.263 | 0.155 |
| ComplEx (Trouillon et al., 2016) | 0.941 | 0.947 | 0.936 | **0.936** | 0.440 | 0.510 | 0.460 | 0.410 | 0.247 | 0.428 | 0.275 | 0.158 |
| ConvE (Dettmers et al., 2018) | **0.943** | **0.956** | **0.946** | 0.935 | 0.430 | 0.520 | 0.440 | 0.400 | 0.325 | 0.501 | 0.356 | 0.237 |
| NeuralLP (Das et al., 2017) | — | — | — | — | **0.463** | **0.657** | **0.468** | 0.376 | 0.227 | 0.348 | 0.248 | 0.166 |
| MINERVA (Das et al., 2017) | — | — | — | — | 0.448 | 0.513 | 0.456 | **0.413** | 0.293 | 0.456 | 0.329 | 0.217 |
| DistMult (emb. size $d = 100$) | 0.782 | 0.931 | 0.910 | 0.658 | 0.411 | 0.463 | 0.423 | 0.382 | **0.214** | **0.402** | **0.240** | **0.127** |
| ComplEx (emb. size $d = 100$) | **0.923** | **0.948** | **0.941** | **0.904** | 0.420 | 0.469 | 0.431 | 0.395 | 0.206 | 0.373 | 0.222 | 0.126 |
| NaNTP | 0.539 | 0.832 | 0.703 | 0.349 | 0.137 | 0.250 | 0.148 | 0.083 | 0.197 | 0.330 | 0.209 | **0.131** |
| NaNTP+Text | | | | | | | | | **0.198** | **0.335** | **0.210** | **0.131** |
| NaNTP+Text+Attention | **0.769** | **0.937** | **0.884** | **0.649** | **0.398** | **0.432** | **0.402** | **0.377** | 0.176 | 0.310 | 0.185 | 0.110 |

Table 2: Explanations, in terms of rules and supporting facts, for the queries in the validation set of WN18 and WN18RR provided by NaNTPs by looking at the proof paths yielding the largest proof scores.

| | Query | Score $S_\rho$ | Proofs / Explanations |
|---|---|---|---|
| **WN18** | part_of(CONGO.N.03, AFRICA.N.01) | 0.995 | part_of(X, Y) :- has_part(Y, X)
has_part(AFRICA.N.01, CONGO.N.03) |
| | | 0.787 | part_of(X, Y) :- instance_hyponym(Y, X)
instance_hyponym(AFRICAN_COUNTRY.N.01, CONGO.N.03) |
| | hyponym(EXTINGUISH.V.04, DECOUPLE.V.03) | 0.987 | hyponym(X, Y) :- hypernym(Y, X)
hypernym(DECOUPLE.V.03, EXTINGUISH.V.04) |
| | | 0.920 | hypernym(SNUFF_OUT.V.01, EXTINGUISH.V.04) |
| | part_of(PITUITARY.N.01, DIENCEPHALON.N.01) | 0.995 | has_part(DIENCEPHALON.N.01, PITUITARY.N.01) |
| | has_part(TEXAS.N.01, ODESSA.N.02) | 0.961 | has_part(X, Y) :- part_of(Y, X)
part_of(ODESSA.N.02, TEXAS.N.01) |
| | hyponym(SKELETAL_MUSCLE, ARTICULAR_MUSCLE) | 0.987 | hypernym(ARTICULAR_MUSCLE, SKELETAL_MUSCLE) |
| | deriv_related_form(REWRITE, REWRITING) | 0.809 | deriv_related_form(X, Y) :- hypernym(Y, X)
hypernym(REVISE, REWRITE) |
| **WN18RR** | also_see(TRUE.A.01, FAITHFUL.A.01) | 0.962 | also_see(X, Y) :- also_see(Y, X)
also_see(FAITHFUL.A.01, TRUE.A.01) |
| | | 0.590 | also_see(CONSTANT.A.02, FAITHFUL.A.01) |
| | also_see(GOOD.A.03, VIRTUOUS.A.01) | 0.962 | also_see(VIRTUOUS.A.01, GOOD.A.03) |
| | | 0.702 | also_see(RIGHTEOUS.A.01, VIRTUOUS.A.01) |
| | instance_hypernym(CHAPLIN, FILM_MAKER) | 0.812 | instance_hypernym(CHAPLIN, COMEDIAN) |

for Countries. Results are reported in terms of Area Under the Precision-Recall Curve (Davis & Goadrich, 2006) (AUC-PR), Mean Reciprocal Rank (MRR), and HITS@$m$ (Bordes et al., 2013). All datasets are described in detail in Appendix B.

**Baselines.** We compare NaNTPs with NTPs on benchmark datasets, and with DistMult (Yang et al., 2015) and ComplEx (Trouillon et al., 2016), two state-of-the-art Neural Link Predictors used for identifying missing facts in potentially very large KBs, on WordNet and FreeBase. For computing likelihoods of facts, DistMult and ComplEx embed each entity and relation type in a $d$-dimensional embedding space and use a differentiable scoring function based on the embeddings corresponding to the entity and relation of a fact. Embedding representations and scoring function parameters are learned jointly by minimising a KB reconstruction error.

Note that, while the complexity of scoring a triple in DistMult and ComplEx is $\mathcal{O}(1)$—they only need the embeddings of the symbols within a fact for computing the ranking score—instead in our model, it is $\mathcal{O}(\log |\mathfrak{K}|)$. For such a reason, instead of performing a full hyperparameter search, we fix some of the hyperparameters—*i.e.* we fix the embedding size $d = 100$, and train them for 100 epochs—and report results using these hyperparameters for NaNTPs.

For the sake of comparison, we also train ComplEx and DistMult while fixing $d = 100$ and the number of training epochs to 100, similarly to NaNTPs.

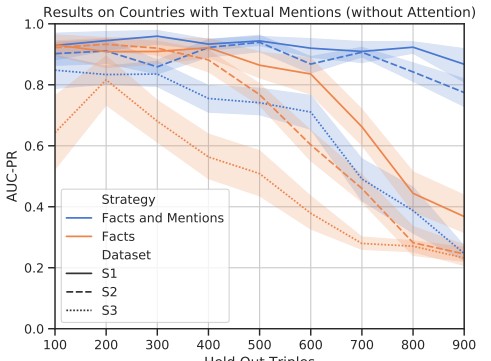 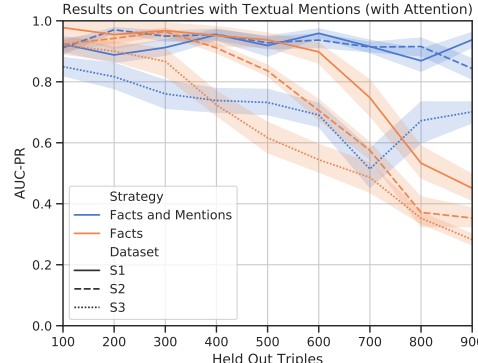

Figure 2: Given the Countries dataset, we replaced a varying number of training triples with mentions (see Appendix B.1.2 for details) and integrated the mentions using two different strategies: by encoding the mentions using the encoder introduced in Section 4 (*Facts and Mentions*) and by simply adding them to the KB (*Facts*).Experiments were conducted with the attention mechanism proposed in Section 3 (right) and the standard rule-learning procedure (left), each with 10 different random seeds. We can see that, on each of the datasets, using the encoder yields consistently better AUC-PR values than simply adding the mentions to the KB.

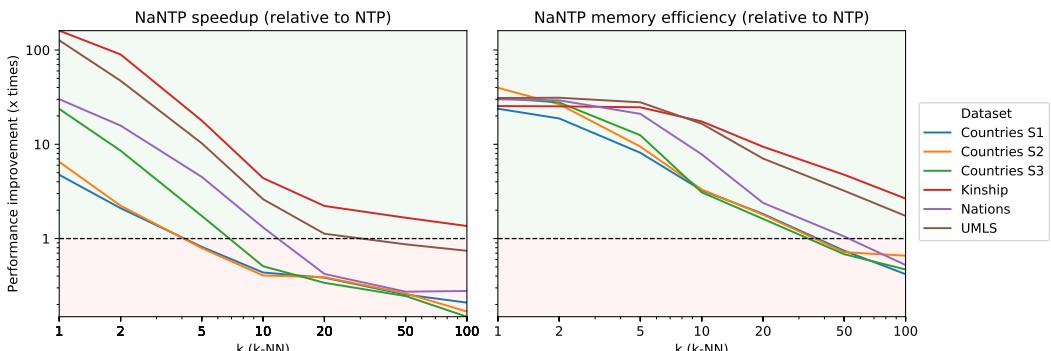

Figure 3: Run-time and memory performance of NaNTP in comparison with NTP. Run-time speedup calculated as the ratio of examples per second of NaNTP and NTP. Memory efficiency calculated as a ratio of the memory use of NTP and NaNTP. Dashed line denotes equal performance – above it (green) NaNTP performs better, below it (red) performs worse.

## 6.2 SCALABILITY EXPERIMENTS

**Evaluation Performance Comparison** In order to verify the correctness of our approximation, we compare the evaluation performance of NaNTP and NTP on the set of benchmark datasets presented in Rocktäschel & Riedel (2017). Results, presented in Table 3 show that NaNTP achieves on par or better results than NTP, consistently through benchmark datasets. Please note that results reported in Rocktäschel & Riedel (2017) were calculated with an incorrect ranking function, which caused them to report artificially better ranking results.[2]

**Run-Time Performance comparison** To assess the run-time gains of NaNTP, we compare it to NTP with respect to time and memory performance during training. In our experiments, we vary the $n$ of the ANNS approximation to assess the computational demands by increasing $n$. First, we compare the average number of examples (queries) per second by running 10 training batches with a maximum batch to fit the memory of NVIDIA GeForce GTX 1080 Ti, for all models. Second, we

---

[2]In their implementation, if several facts have the same score, the ranking function assigns them the same (best) rank, which artificially inflated their results.

Table 3: Comparison of NaNTPs and NTPs on benchmark datasets. Double asterisk (**) denotes the performance of NTP reevaluated with the correct evaluation function (see the note in Section 6.2). Results for DistMult, ComplEx, ConvE, NeuralLP, and MINERVA are from Das et al. (2017).

| | | Models | | | | | | |
|---|---|---|---|---|---|---|---|---|
| Datasets | Metrics | NTP** | NaNTP | | | ComplEx | NeuralLP | MINERVA |
| | | | k=1 | k=2 | k=5 | | | |
| Countries | S1 | $90.83 \pm 15.4$ | $\mathbf{99.20 \pm 1.19}$ | $97.34 \pm 3.86$ | $96.37 \pm 4.23$ | $99.37 \pm 0.4$ | $\mathbf{100.0 \pm 0.0}$ | $\mathbf{100.0 \pm 0.0}$ |
| | S2 AUC-PR | $87.40 \pm 11.7$ | $\mathbf{93.48 \pm 3.29}$ | $88.48 \pm 5.87$ | $83.56 \pm 7.78$ | $87.95 \pm 2.8$ | $75.1 \pm 0.3$ | $92.36 \pm 2.41$ |
| | S3 | $56.68 \pm 17.6$ | $\mathbf{85.24 \pm 5.83}$ | $82.18 \pm 9.11$ | $72.68 \pm 13.4$ | $48.44 \pm 6.3$ | $92.2 \pm 0.2$ | $\mathbf{95.10 \pm 1.20}$ |
| Kinship | MRR | 0.35 | $0.51 \pm 0.04$ | $\mathbf{0.66 \pm 0.03}$ | $\mathbf{0.67 \pm 0.02}$ | **0.838** | 0.619 | 0.720 |
| | HITS@1 | 0.24 | $0.37 \pm 0.04$ | $\mathbf{0.51 \pm 0.04}$ | $\mathbf{0.52 \pm 0.02}$ | **0.754** | 0.475 | 0.605 |
| | HITS@3 | 0.37 | $0.57 \pm 0.05$ | $\mathbf{0.78 \pm 0.03}$ | $\mathbf{0.78 \pm 0.02}$ | **0.910** | 0.707 | 0.812 |
| | HITS@10 | 0.57 | $0.82 \pm 0.04$ | $\mathbf{0.94 \pm 0.02}$ | $\mathbf{0.95 \pm 0.00}$ | **0.980** | 0.912 | 0.924 |
| Nations | MRR | 0.61 | $\mathbf{0.66 \pm 0.03}$ | $0.59 \pm 0.02$ | $0.54 \pm 0.05$ | — | — | — |
| | HITS@1 | 0.45 | $\mathbf{0.51 \pm 0.05}$ | $0.41 \pm 0.03$ | $0.36 \pm 0.07$ | — | — | — |
| | HITS@3 | 0.73 | $\mathbf{0.77 \pm 0.03}$ | $0.71 \pm 0.02$ | $0.65 \pm 0.06$ | — | — | — |
| | HITS@10 | 0.87 | $\mathbf{0.99 \pm 0.00}$ | $0.98 \pm 0.01$ | $0.98 \pm 0.01$ | — | — | — |
| UMLS | MRR | **0.80** | $0.63 \pm 0.04$ | $\mathbf{0.80 \pm 0.02}$ | $\mathbf{0.80 \pm 0.02}$ | **0.894** | 0.778 | 0.825 |
| | HITS@1 | **0.70** | $0.49 \pm 0.05$ | $\mathbf{0.68 \pm 0.02}$ | $\mathbf{0.68 \pm 0.03}$ | **0.823** | 0.643 | 0.728 |
| | HITS@3 | 0.88 | $0.73 \pm 0.04$ | $\mathbf{0.91 \pm 0.01}$ | $\mathbf{0.90 \pm 0.02}$ | **0.962** | 0.869 | 0.900 |
| | HITS@10 | 0.95 | $0.89 \pm 0.02$ | $\mathbf{0.98 \pm 0.01}$ | $\mathbf{0.97 \pm 0.01}$ | **0.995** | 0.962 | 0.968 |

compare the maximum memory usage of both models on a CPU, over 10 training batches with same batch sizes. The comparison is done on a CPU to ensure that we include the size of the ANNS index in NaNTP measures and as a fail-safe, in case the model does not fit on the GPU memory.

The results, presented in Figure 3, demonstrate that, compared to NTP, NaNTP is considerably more time and memory efficiency. In particular, we observe that NaNTP yields significant speedups of an order of magnitude for smaller datasets (Countries S1 and S2), and more than two orders of magnitude for larger datasets (Kinship and Nations). Interestingly, with the increased size of the dataset, NaNTP consistently achieves higher speedups, when compared to NTP. Similarly, NaNTP is more memory efficient, with savings bigger than an order of magnitude, making them readily applicable to larger datasets, even when augmented with textual surface forms.

## 6.3 RESULTS ON COUNTRIES, UMLS, AND NATIONS

**Experiments with Generated Mentions.** For evaluating different strategies of integrating textual surface patterns, in the form of mentions, in NTPs, we proceeded as follows. We replaced a varying number of training set triples from each of the Countries S1-3 datasets with human-generated textual mentions. For instance, the fact `neighbourOf(`UK, IRELAND`)` may be replaced by the mention "UK `is neighbouring with` IRELAND".

Then, we evaluate two ways of integrating textual mentions in NaNTPs, either by i) adding them as facts to the KB, or by ii) parsing the mention by means of an encoder, as described in Section 4. The results, presented in Fig. 2, testify that the proposed encoding module yields consistent improvements of the ranking accuracy in comparison to simply adding the mentions as facts to the KB. This is particularly obvious in cases where the number of held-out facts is higher, implying that the added mentions can replace a large missing number of original facts in the KB.

**Explanations Involving Mentions.** NaNTPs are extremely efficient at learning rules involving both *logic atoms and textual mentions*. For instance, by analysing the learned models and their explanations, we can see that NaNTPs learn patterns such as:

```
neighborOf(X, Y) :–  neighborOf(Y, X)
neighborOf(X, Y) :–  "E₁ was a neighbor of E₂"(Y, X)
neighborOf(X, Y) :–  "E₁ is a neighboring state to E₂"(Y, X)
 locatedIn(X, Y) :–  "E₁ was a neighboring state to E₂"(X, Z), "E₁ was located in E₂"(Z, Y)
 locatedIn(X, Y) :–  "E₁ can be found in E₂"(X, Z), "E₁ is located in E₂"(Z, Y)
```

where $E_1$ and $E_2$ denote the position of the entities in the text surface patterns, and leverage them during their reasoning process, providing human-readable explanations for a given prediction.

## 6.4 RESULTS ON WORDNET AND FREEBASE

Link prediction results are summarised in Table 1, while Table 2 shows a sample of explanations for the facts in the validation set of WN18 and WN18RR provided by NaNTPs by analysing the proof paths associated with the largest proof scores. We can see that NaNTPs is capable of learning rules, such as $\texttt{has\_part}(X, Y) :- \texttt{part\_of}(Y, X)$, and $\texttt{hyponym}(X, Y) :- \texttt{hypernym}(Y, X)$.

Interestingly, it is also able to find an alternative, non-trivial explanations for a given fact, based on the similarity between entity representations. For instance, it can explain that CONGO is part of AFRICA by leveraging the similarity between AFRICA and AFRICAN_ COUNTRY, and the fact that the latter is a hyponym of CONGO. It is also able to explain that CHAPLIN is a FILM_MAKER by leveraging the prior knowledge that CHAPLIN is a COMEDIAN, and the similarity between FILM_MAKER and COMEDIAN.

## 6.5 THE EFFECT OF ATTENTION

We analysed the effect of using attention for rule learning, introduced in Section 3, on NaNTP's accuracy, on both the benchmark datasets—outlined in Table 6—and WordNet—outlined in Table 7.

Table 6 shows that using attention in NaNTP for learning rule representations yields higher average ranking accuracy and lower variance on Countries S1-3 and Kinship, while yielding comparable results to not using attention on Nations and UMLS. This is consistent with the observation in Fig. 2, where NaNTPs with attention yield better performance on Countries S1-3.

Results in Table 7 show the results of ablations on two large datasets derived from WordNet, namely WN18 and WN18RR. For these two datasets, attention for learning rules greatly increases the ranking accuracy. For instance Hits@10 increases from $83.2\%$ to $93.7\%$ in the case of WN18, and from $25\%$ to $43.2\%$ in the case of WN18RR. Please note that state-of-the-art Neural Link Predictors such as ComplEx (Trouillon et al., 2016) still yield an Hits@10 lower than $95\%$ on WN18: this shows that WN18 yields results on par with other classes of models, while providing explanations for each prediction (as shown in Section 6.4). Note that Neural Link Predictors are a class of model that was investigated for more than a decade now (Paccanaro & Hinton, 2001; Bordes et al., 2013).

Similarly, MRR increases from $0.539$ to $0.769$ in the case of WN18, and from $0.137$ to $0.398$ in the case of WN18RR. An explanation for this phenomenon is that using attention drastically reduces the number of parameters required to learn each of the rule predicates from $100$ to $18$ in the case of WN18, and to $11$ in the case of WN18RR, introducing an inductive bias that reveals being extremely beneficial in terms of ranking accuracy. We can also note that, for FB15k-237, attention did not improve the ranking accuracy. An explanation is that this dataset is very high relational (237 relation types), and using attention actually *increased* the number of parameters to be learned.

## 7 CONCLUSION

NTPs combine the strengths of rule-based and neural models but, so far, they were unable to reason over large KBs, and therefore over natural language.

In this paper, we proposed NaNTPs that utilise ANNS and attention as a solution to scaling issues of NTP. By efficiently considering only the subset of proof paths associated with the highest proof scores during the construction of a dynamic computation graph, NaNTPs yield drastic speedups and memory efficiency, while yielding the same or a better predictive accuracy than NTPs. This enables application of NaNTPs to mixed KB and natural language data by embedding logic atoms and textual mentions in a joint embedding space.

Albeit results are still slightly lower than those yielded by state-of-the-art Neural Link Predictors on large datasets, NaNTPs is interpretable and is able to provide explanations of its reasoning at scale.

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

# A  NEURAL THEOREM PROVING

In NTPs, the neural network structure is built recursively, and its construction is defined in terms of *modules* similarly to dynamic neural module networks (Andreas et al., 2016). Given a goal, a KB, and a current proof state as inputs, each module produces a list of new proof states, *i.e.*, neural networks representing partial proof success scores and variable substitutions. In the following we briefly overview the modules constituting NTPs.

## A.1  UNIFICATION MODULE

In backward chaining, unification between two logic atoms is used for checking whether they can represent the same structure. In discrete unification, non-variable symbols are checked for equality, and the proof fails if two symbols differ. Rather than comparing symbols, NTPs compare their *embedding representations* by means of an end-to-end differentiable similarity function, such as a RBF kernel. This allows matching different symbols with similar semantics, such as relations like `locatedIn` and `situatedIn`.

Unification is carried out by the `unify` operator, which updates a substitution set $S$, and creates a neural network for comparing the vector representations of non-variable symbols in two sequences of terms. The signature of `unify` is $\mathcal{L} \times \mathcal{L} \times \mathcal{S} \to \mathcal{S}$, where $\mathcal{L}$ is the domain of lists of terms: it takes two atoms, represented as lists of terms, and an upstream proof state, and returns a new proof state $S'$.

More formally, let H and G denote two lists of terms, each denoting a logical atom, such as $[\texttt{p}, \texttt{A}, \texttt{B}]$ and $[\texttt{q}, \texttt{X}, \texttt{Y}]$ for respectively denoting the atoms $\texttt{p}(\texttt{A}, \texttt{B})$ and $\texttt{q}(\texttt{X}, \texttt{Y})$. Given a *proof state* $S = (S_\psi, S_\rho)$, where $S_\psi$ and $S_\rho$ respectively denote a *substitution set* and a *proof score*, unification is computed as follows:

1. $\texttt{unify}_{\boldsymbol{\theta}}([\,], [\,], S) = S$
2. $\texttt{unify}_{\boldsymbol{\theta}}([\,], \text{G}, S) = \texttt{FAIL}$

3. $\texttt{unify}_{\boldsymbol{\theta}}(\text{H}, [\,], S) = \texttt{FAIL}$
4. $\texttt{unify}_{\boldsymbol{\theta}}(h :: \text{H}, g :: \text{G}, S) = \texttt{unify}_{\boldsymbol{\theta}}(\text{H}, \text{G}, S')$
   with $S' = (S'_\psi, S'_\rho)$ where:

$$S'_\psi = S_\psi \cup \left\{ \begin{array}{ll} \{h/g\} & \text{if } h \in \mathcal{V} \\ \{g/h\} & \text{if } g \in \mathcal{V}, h \notin \mathcal{V} \\ \varnothing & \text{otherwise} \end{array} \right\}, \quad S'_\rho = \min \left( S_\rho, \left\{ \begin{array}{ll} k\left(\boldsymbol{\theta}_{h:}, \boldsymbol{\theta}_{g:}\right) & \text{if } h \notin \mathcal{V}, g \notin \mathcal{V} \\ 1 & \text{otherwise} \end{array} \right\} \right)$$

Here, $S'$ refers to the new proof state, $\mathcal{V}$ refers to the set of variable symbols, $h/g$ is a substitution from the variable symbol $h$ to the symbol $g$, and $\boldsymbol{\theta}_{g:}$ denotes the embedding look-up of the non-variable symbol with index $g$.

For example, given two atoms H = $[\texttt{locatedIn}, \texttt{LONDON}, \texttt{UK}]$ and G = $[\texttt{situatedIn}, \texttt{X}, \texttt{Y}]$, the result of $\texttt{unify}_{\boldsymbol{\theta}}(\text{H}, \text{G}, (\varnothing, 1))$ with $S = (\varnothing, 1)$ will be a new substitution set $S' = (S'_\psi, S'_\rho)$ where $S'_\psi = \{\texttt{X}/\texttt{LONDON}, \texttt{Y}/\texttt{UK}\}$ and $S'_\rho = \min\left(1, k\left(\boldsymbol{\theta}_{\texttt{locatedIn}:}, \boldsymbol{\theta}_{\texttt{situatedIn}:}\right)\right)$

## A.2  OR MODULE

Given a goal G, the `or` module unifies G with all facts and rules in a KB: for each rule H :– B $\in \mathfrak{K}$, after unifying G with the head H, it also attempts to prove the atoms in the body H by invoking the `and` module. The signature of `or` is $\mathcal{L} \times \mathbb{N} \times \mathcal{S} \to \mathcal{S}^N$, where $\mathcal{L}$ is the domain of goal atoms, the second argument specifies the maximum proof depth, and $N$ denotes the number of possible output proof states. This operator is implemented as follows:

$$\texttt{or}_{\boldsymbol{\theta}}^{\mathfrak{K}}(\text{G}, d, S) = [S' \mid S' \in \texttt{and}_{\boldsymbol{\theta}}^{\mathfrak{K}}(\text{B}, d, \texttt{unify}_{\boldsymbol{\theta}}(\text{H}, \text{G}, S)), \text{H} :\!- \text{B} \in \mathfrak{K}] \tag{7}$$

where H :– B denotes a rule in a given KB $\mathfrak{K}$ with a head atom H and a list of body atoms B.

For example, given a goal G = $[\texttt{situatedIn}, \texttt{Q}, \texttt{UK}]$ and a rule H :– B with H = $[\texttt{locatedIn}, \texttt{X}, \texttt{Y}]$ and B = $[[\texttt{locatedIn}, \texttt{X}, \texttt{Z}], [\texttt{locatedIn}, \texttt{Z}, \texttt{Y}]]$, the model would instantiate an `and` sub-module as follows:

$$\texttt{or}_{\boldsymbol{\theta}}^{\mathfrak{K}}(\text{G}, d, S) = [S' | S' \in \texttt{and}_{\boldsymbol{\theta}}^{\mathfrak{K}}(\text{B}, d, (\{\texttt{X}/\texttt{Q}, \texttt{Y}/\texttt{UK}\}, \hat{S}_\rho)), \ldots]$$

by first unifying the goal G with the head of the rule H, and then proving the sub-goals in the body of the rule by using and.

## A.3 AND MODULE

The and module recursively tries to prove a list of sub-goals, by invoking the or module. Its signature is $\mathcal{L} \times \mathbb{N} \times \mathcal{S} \to \mathcal{S}^N$, where $\mathcal{L}$ is the domain of lists of atoms, and $N$ is the number of possible output proof states for a list of atoms with a known structure and a provided KB. This module is implemented as:

1. $\text{and}_{\boldsymbol{\theta}}^{\mathfrak{K}}(\_,\_,\text{FAIL}) = \text{and}_{\boldsymbol{\theta}}^{\mathfrak{K}}(\_,0,\_) = \text{FAIL}$     2. $\text{and}_{\boldsymbol{\theta}}^{\mathfrak{K}}([\,],\_,S) = S$
3. $\text{and}_{\boldsymbol{\theta}}^{\mathfrak{K}}(\text{G}:\mathbb{G},d,S) = [S'' \mid S'' \in \text{and}_{\boldsymbol{\theta}}^{\mathfrak{K}}(\mathbb{G},d,S'), S' \in \text{or}_{\boldsymbol{\theta}}^{\mathfrak{K}}(\text{substitute}(\text{G},S_{\psi}),d-1,S)]$

where substitute is an auxiliary function that applies substitutions to variables in an atom whenever possible. Line 3 defines the recursion—the first sub-goal is proven by instantiating an or module after substitutions are applied, and every resulting proof state is used for proving the remaining sub-goals by instantiating an and module.

## A.4 PROOF AGGREGATION

Finally, NTPs define the overall success score of proving a goal G using a KB $\mathfrak{K}$ with parameters $\boldsymbol{\theta}$ as:

$$\text{ntp}_{\boldsymbol{\theta}}^{\mathfrak{K}}(\text{G},d) = \operatorname*{arg\,max}_{\substack{S \in \text{or}_{\boldsymbol{\theta}}^{\mathfrak{K}}(\text{G},d,(\varnothing,1)) \\ S \neq \text{FAIL}}} S_{\rho}$$

where $d$ is a predefined maximum proof depth, and the initial proof state is set to $(\varnothing, 1)$, denoting an empty substitution set and a proof success score of 1.

## A.5 TRAINING

The model is then trained using a Leave-One-Out cross-entropy loss—by removing one fact from the KB, and predicting its score. Since this procedure only generates positive examples, negative examples are generated by corrupting positive examples, by randomly changing the entities (Nickel et al., 2016). We refer to Rocktäschel & Riedel (2017) for more information on the training procedure.

## A.6 RULE LEARNING

NTPs can be used for learning *interpretable rules* from data, getting a deeper understanding of the domain. For example, consider the rule H :– B, with H $= [\text{grandfatherOf}, \text{X}, \text{Y}]$ and B $= [[\text{fatherOf}, \text{X}, \text{Z}], [\text{parentOf}, \text{Z}, \text{Y}]]$. Rocktäschel & Riedel (2017) show that it is possible to learn this rule from data by specifying a *rule template* H :– B, with H $= [\boldsymbol{\theta}_{p:}, \text{X}, \text{Y}]$ and B $= [[\boldsymbol{\theta}_{q:}, \text{X}, \text{Z}], [\boldsymbol{\theta}_{r:}, \text{Z}, \text{Y}]]$, where $\boldsymbol{\theta}_{p:}, \boldsymbol{\theta}_{q:}, \boldsymbol{\theta}_{r:} \in \mathbb{R}^k$. The parameters $\boldsymbol{\theta}_{p:}, \boldsymbol{\theta}_{q:}, \boldsymbol{\theta}_{r:}$ can be *learned from data*, and decoded by searching the closest representation of known predicates.

# B DATASETS

We run experiments on the following datasets—also outlined in Table 5—and report results in terms of Area Under the Precision-Recall Curve (Davis & Goadrich, 2006) (AUC-PR), MRR, and HITS@$m$ (Bordes et al., 2013).

## B.1 COUNTRIES, UMLS, NATIONS

### B.1.1 COUNTRIES

Countries is a dataset introduced by Bouchard et al. (2015) for testing reasoning capabilities of neural link prediction models. It consists of 244 countries, 5 regions (*e.g.* EUROPE), 23 sub-regions (*e.g.* WESTERN EUROPE, NORTH AMERICA), and 1158 facts about the neighbourhood of countries, and the location of countries and sub-regions. As in Rocktäschel & Riedel (2017), we randomly split

| Predicate Name | Mentions |
|---|---|
| locatedIn$(a, b)$ | $a$ is located in $b$, $a$ is situated in $b$, $a$ is placed in $b$, $a$ is positioned in $b$, $a$ is sited in $b$, $a$ is currently in $b$, $a$ can be found in $b$, $a$ is still in $b$, $a$ is localized in $b$, $a$ is present in $b$, $a$ is contained in $b$, $a$ is found in $b$, $a$ was located in $b$, $a$ was situated in $b$, $a$ was placed in $b$, $a$ was positioned in $b$, $a$ was sited in $b$, $a$ was currently in $b$, $a$ used to be found in $b$, $a$ was still in $b$, $a$ was localized in $b$, $a$ was present in $b$, $a$ was contained in $b$, $a$ was found in $b$ |
| neighborOf$(a, b)$ | $a$ is adjacent to $b$, $a$ borders with $b$, $a$ is butted against $b$, $a$ neighbours $b$, $a$ is a neighbor of $b$, $a$ is a neighboring country of $b$, $a$ is a neighboring state to $b$, $a$ was adjacent to $b$, $a$ borders $b$, $a$ was butted against $b$, $a$ neighbours with $b$, $a$ was a neighbor of $b$, $a$ was a neighboring country of $b$, $a$ was a neighboring state to $b$ |

Table 4: Mentions used for replacing a varying number of training triples in the Countries S1, S2, and S3 datasets.

countries into a training set of 204 countries (train), a development set of 20 countries (validation), and a test set of 20 countries (test), such that every validation and test country has at least one neighbour in the training set. Subsequently, three different task datasets are created, namely **S1**, **S2**, and **S3**. For all tasks, the goal is to predict locatedIn$(c, r)$ for every test country $c$ and all five regions $r$, but the access to training atoms in the KB varies.

**S1:** All ground atoms locatedIn$(c, r)$, where $c$ is a test country and $r$ is a region, are removed from the KB. Since information about the sub-region of test countries is still contained in the KB, this task can be solved by using the transitivity rule locatedIn$(X, Y)$ :– locatedIn$(X, Z)$, locatedIn$(Z, Y)$.

**S2:** In addition to **S1**, all ground atoms locatedIn$(c, s)$ are removed where $c$ is a test country and $s$ is a sub-region. The location of countries in the test set needs to be inferred from the location of its neighbouring countries: locatedIn$(X, Y)$ :– neighborOf$(X, Z)$, locatedIn$(Z, Y)$. This task is more difficult than **S1**, as neighbouring countries might not be in the same region, so the rule above will not always hold.

**S3:** In addition to **S2**, also all ground atoms locatedIn$(c, r)$ are removed where $r$ is a region and $c$ is a country from the training set training that has a country from the validation or test sets as a neighbour. The location of test countries can for instance be inferred using the rule locatedIn$(X, Y)$ :– neighborOf$(X, Z)$, neighborOf$(Z, W)$, locatedIn$(W, Y)$.

### B.1.2    COUNTRIES WITH MENTIONS

We generated a set of variants of Countries S1, S2, and S3, by randomly replacing a varying number of training set triples with mentions. The employed mentions are outlined in Table 4.

### B.1.3    NATIONS AND UMLS

Furthermore, we consider the Nations, and the Unified Medical Language System (UMLS) datasets (Kok & Domingos, 2007). UMLS contains 49 predicates, 135 constants and 6529 true facts, while Nations contains 56 binary predicates, 111 unary predicates, 14 constants and 2565 true facts. We follow the protocol used by Rocktäschel & Riedel (2017) and split every dataset into training, development, and test facts, with a $80\%/10\%/10\%$ ratio. For evaluation, we take a test fact and corrupt its first and second argument in all possible ways such that the corrupted fact is not in the original KB. Subsequently, we predict a ranking of the test fact and its corruptions to calculate MRR and HITS@$m$.

Note that neither Countries nor UMLS and Nations have mentions. For such a reason, for each of these datasets, we generated one equivalent mention—using a natural language sentence rather than a predicate name—and replaced a varying amount of training set triples with equivalent mentions.

Table 5: Dataset statistics.

| Dataset Name | #Rel. | #Entities | #Train | #Validation | #Test | #Mentions |
|---|---|---|---|---|---|---|
| **FB15k-237.E** (Toutanova et al., 2015) | 237 | 27,395 | 272,115 | 17,535 | 20,466 | 3,978,014 |
| **WN18** (Bordes et al., 2013) | 18 | 40,943 | 141,442 | 5,000 | 5,000 | — |
| **WN18RR** (Dettmers et al., 2018) | 11 | 40,943 | 86,835 | 3,034 | 3,134 | — |

### B.2 WORDNET

WN18 (Bordes et al., 2013) is a subset of WordNet (Miller, 1995), a lexical KB for the English language, where entities correspond to word senses and relationships define lexical relations between them. We also consider WN18RR (Dettmers et al., 2018), a dataset derived from WN18 where predicting missing links is sensibly harder.

### B.3 FREEBASE

For evaluating the impact of also using mentions, we use the FB15k-237.E dataset (Toutanova & Chen, 2015; Toutanova et al., 2015), a subset of FB15k (Bordes et al., 2013) that excludes redundant relations and direct training links for held-out triples, with the goal of making the task more realistic. Textual relations for FB15k-237.E are extracted from 200 million sentences in the ClueWeb12 corpus, coupled with Freebase mention annotations (Gabrilovich et al., 2013), and include textual links of all co-occurring entities from the KB set. After pruning [3], there are 2.7 million unique textual relations that are added to the KB. The number of relations and triples in the training, validation, and test portions of the data are given in Table 5. In particular, FB15k-237.E denotes the FB15k-237 dataset augmented with textual relations proposed by Toutanova et al. (2015).

### B.4 WORDNET AND FREEBASE STATISTICS

In order to grasp the magnitude of WN18, WN18RR and FB15k-237.E datasets, we provide their basic statistics in Table 5.

## C ADDITIONAL EXPERIMENTS

### C.1 ABLATION STUDIES

The effect of attention over rules to this framework is quantised by two ablation studies, on benchmark datasets, in Table 6, and on the large datasets, in Table 7 shows attention achieving higher or on-par performance with NaNTP without attention. Table 6, however, reports that NaNTP with attention significantly outperforms NaNTP without it on WordNet datasets.

### C.2 ANNS VS. EXACT NNS VS. RANDOM SELECTION

In order to analyse the impact of using ANNS as a choice of heuristic, we ran additional experiments on the baseline datasets. In particular, we compared ANNS to exact nearest neighbours search, since ANNS models may not return the exact nearest neighbours. We also compared ANNS to random neighbour selection, for analysing the behaviour of the model with random neighbourhood choices.

Results are outlined in Table 8: they show that the random neighbour, as expected, yield sensibly worse ranking results in comparison with ANNS.

A surprising exception is Nations, where ranking results were apparently higher in comparison with UMLS and Kinship: a possible explanation is that Nations only contains 14 entities, so the random neighbourhood can sometimes correspond to the exact neighbourhood.

---

[3]The full set of 37 million textual patterns connecting the entity pairs of interest was pruned based on the count of patterns and their tri-grams, and their precision in indicating that entity pairs have KB relations.

Table 6: Ablation of attention over relations on NaNTPs, on benchmark datasets.

| Datasets | Metrics | | NaNTP | | | | | |
|---|---|---|---|---|---|---|---|---|
| | | **Standard** | | | **Attention** | | | |
| | | k=1 | k=2 | k=5 | k=1 | k=2 | k=5 | |
| **Countries** | S1 | 94.93 ± 7.58 | 96.58 ± 4.41 | 94.01 ± 3.6 | **99.20 ± 1.19** | 97.34 ± 3.86 | 96.37 ± 4.23 | |
| | S2 AUC-PR | 90.27 ± 4.53 | 81.27 ± 14.44 | 76.9 ± 14.66 | **93.48 ± 3.29** | 88.48 ± 5.87 | 83.56 ± 7.78 | |
| | S3 | 84.16 ± 8.31 | **85.69 ± 9.56** | 80.9 ± 10.12 | **85.24 ± 5.83** | 82.18 ± 9.11 | 72.68 ± 13.4 | |
| **Kinship** | MRR | 0.56 ± 0.03 | 0.61 ± 0.04 | 0.57 ± 0.05 | 0.51 ± 0.04 | **0.66 ± 0.03** | **0.67 ± 0.02** | |
| | HITS@1 | 0.41 ± 0.03 | 0.45 ± 0.05 | 0.41 ± 0.05 | 0.37 ± 0.04 | **0.51 ± 0.04** | **0.52 ± 0.02** | |
| | HITS@3 | 0.64 ± 0.03 | 0.72 ± 0.04 | 0.67 ± 0.05 | 0.57 ± 0.05 | **0.78 ± 0.03** | **0.78 ± 0.02** | |
| | HITS@10 | 0.87 ± 0.02 | 0.92 ± 0.02 | 0.88 ± 0.03 | 0.82 ± 0.04 | **0.94 ± 0.02** | **0.95 ± 0.00** | |
| **Nations** | MRR | 0.63 ± 0.04 | **0.65 ± 0.03** | 0.60 ± 0.05 | **0.66 ± 0.03** | 0.59 ± 0.02 | 0.54 ± 0.05 | |
| | HITS@1 | 0.46 ± 0.05 | **0.48 ± 0.03** | 0.42 ± 0.07 | **0.51 ± 0.05** | 0.41 ± 0.03 | 0.36 ± 0.07 | |
| | HITS@3 | 0.74 ± 0.04 | **0.76 ± 0.03** | 0.72 ± 0.04 | **0.77 ± 0.03** | 0.71 ± 0.02 | 0.65 ± 0.06 | |
| | HITS@10 | **0.99 ± 0.01** | **0.99 ± 0.00** | 0.98 ± 0.01 | **0.99 ± 0.00** | 0.98 ± 0.01 | 0.98 ± 0.01 | |
| **UMLS** | MRR | 0.57 ± 0.04 | **0.81 ± 0.01** | 0.80 ± 0.02 | 0.63 ± 0.04 | **0.80 ± 0.02** | **0.80 ± 0.02** | |
| | HITS@1 | 0.42 ± 0.04 | **0.68 ± 0.01** | 0.68 ± 0.03 | 0.49 ± 0.05 | **0.68 ± 0.02** | **0.68 ± 0.03** | |
| | HITS@3 | 0.66 ± 0.04 | **0.92 ± 0.01** | 0.91 ± 0.01 | 0.73 ± 0.04 | **0.91 ± 0.01** | **0.90 ± 0.02** | |
| | HITS@10 | 0.85 ± 0.03 | **0.98 ± 0.00** | 0.98 ± 0.01 | 0.89 ± 0.02 | **0.98 ± 0.01** | **0.97 ± 0.01** | |

Table 7: Ablation of attention over relations on NaNTPs, on WN18, WN18RR and FB15k-237.E

| | **WN18** | | | | **WN18RR** | | | | **FB15k-237.E** | | | |
|---|---|---|---|---|---|---|---|---|---|---|---|---|
| | | Hits | | | | Hits | | | | Hits | | |
| | MRR | @10 | @3 | @1 | MRR | @10 | @3 | @1 | MRR | @10 | @3 | @1 |
| DistMult (Yang et al., 2015) | 0.822 | 0.936 | 0.914 | 0.728 | 0.430 | 0.490 | 0.440 | 0.390 | 0.241 | 0.419 | 0.263 | 0.155 |
| ComplEx (Trouillon et al., 2016) | 0.941 | 0.947 | 0.936 | 0.936 | 0.440 | 0.510 | 0.460 | 0.410 | 0.247 | 0.428 | 0.275 | 0.158 |
| ConvE (Dettmers et al., 2018) | 0.943 | 0.520 | 0.440 | 0.400 | 0.430 | 0.520 | 0.440 | 0.400 | 0.325 | 0.501 | 0.356 | 0.237 |
| DistMult ($d = 100$) | 0.782 | 0.931 | 0.910 | 0.658 | 0.411 | 0.463 | 0.423 | 0.382 | 0.214 | 0.402 | 0.240 | 0.127 |
| ComplEx ($d = 100$) | 0.923 | 0.948 | 0.941 | 0.904 | 0.420 | 0.469 | 0.431 | 0.395 | 0.206 | 0.373 | 0.222 | 0.126 |
| NaNTP | 0.539 | 0.832 | 0.703 | 0.349 | 0.137 | 0.250 | 0.148 | 0.083 | 0.197 | 0.330 | 0.209 | **0.131** |
| NaNTP+Text | | | | | | | | | **0.198** | **0.335** | **0.210** | **0.131** |
| NaNTP+Tex+Attention | **0.769** | **0.937** | **0.884** | **0.649** | **0.398** | **0.432** | **0.402** | **0.377** | 0.176 | 0.310 | 0.185 | 0.110 |

We can also observe that ANNS, yield very close ranking results in comparison with Exact NNS, but orders of magnitude faster. This implies that, compared to a costly Exact NNS, ANNS is an optimal choice for a heuristic, since it greatly decreases the computational complexity of the method.

Please note that experiments with Exact NNS were extremely computationally demanding and, for such a reason, we limited the neighbourhood size $k$ to $k = 1$.

Table 8: Performance of NaNTPs with attention (Attention) and without it (Standard) when using the random nearest neighbour, ANNS and exact NNS for $k = 1$, on benchmark datasets.

| Datasets | Metrics | | NaNTP | | | | |
|---|---|---|---|---|---|---|---|
| | | **Random** | | **ANNS** | | **Exact NNS** | |
| | | Standard | Attention | Standard | Attention | Standard | Attention |
| **Countries** | S1 | 40.54 ± 4.85 | 42.88 ± 3.5 | 94.93 ± 7.58 | **99.20 ± 1.19** | 96.65 ± 3.01 | 98.14 ± 3.31 |
| | S2 AUC-PR | 36.02 ± 5.56 | 39.32 ± 4.21 | 90.27 ± 4.53 | **93.48 ± 3.29** | 92.72 ± 4.47 | 91.25 ± 2.18 |
| | S3 | 28.73 ± 2.82 | 38.46 ± 5.81 | 84.16 ± 8.31 | 85.24 ± 5.83 | 85.68 ± 5.46 | **89.12 ± 9.17** |
| **Kinship** | MRR | 0.05 ± 0.00 | 0.06 ± 0.00 | 0.56 ± 0.03 | 0.51 ± 0.04 | **0.58 ± 0.02** | 0.52 ± 0.04 |
| | HITS@1 | 0.01 ± 0.00 | 0.01 ± 0.00 | 0.41 ± 0.03 | 0.37 ± 0.04 | **0.43 ± 0.02** | 0.38 ± 0.04 |
| | HITS@3 | 0.03 ± 0.00 | 0.03 ± 0.00 | 0.64 ± 0.03 | 0.57 ± 0.05 | **0.66 ± 0.03** | 0.58 ± 0.05 |
| | HITS@10 | 0.11 ± 0.00 | 0.11 ± 0.01 | 0.87 ± 0.02 | 0.82 ± 0.04 | **0.89 ± 0.02** | 0.81 ± 0.05 |
| **Nations** | MRR | 0.42 ± 0.01 | 0.42 ± 0.01 | 0.63 ± 0.04 | 0.66 ± 0.03 | **0.70 ± 0.04** | 0.63 ± 0.02 |
| | HITS@1 | 0.19 ± 0.02 | 0.19 ± 0.02 | 0.46 ± 0.05 | 0.51 ± 0.05 | **0.56 ± 0.05** | 0.46 ± 0.03 |
| | HITS@3 | 0.54 ± 0.02 | 0.54 ± 0.01 | 0.74 ± 0.04 | 0.77 ± 0.03 | **0.81 ± 0.03** | 0.74 ± 0.02 |
| | HITS@10 | 0.96 ± 0.01 | 0.96 ± 0.01 | **0.99 ± 0.01** | **0.99 ± 0.00** | **0.99 ± 0.00** | 0.99 ± 0.01 |
| **UMLS** | MRR | 0.13 ± 0.00 | 0.14 ± 0.01 | 0.57 ± 0.04 | **0.63 ± 0.04** | 0.59 ± 0.05 | 0.61 ± 0.05 |
| | HITS@1 | 0.05 ± 0.00 | 0.06 ± 0.01 | 0.42 ± 0.04 | **0.49 ± 0.05** | 0.43 ± 0.05 | 0.46 ± 0.06 |
| | HITS@3 | 0.12 ± 0.01 | 0.13 ± 0.01 | 0.66 ± 0.04 | **0.73 ± 0.04** | 0.68 ± 0.05 | 0.70 ± 0.05 |
| | HITS@10 | 0.27 ± 0.01 | 0.28 ± 0.01 | 0.85 ± 0.03 | **0.89 ± 0.02** | 0.88 ± 0.03 | 0.86 ± 0.04 |

