# OpenReview forum: "Scalable Neural Theorem Proving on Knowledge Bases and Natural Language"
_ICLR.cc/2019/Conference_

### Official Review · AnonReviewer1 · 2018-11-01
**Interesting paper and contributions.**

**Rating:** 5
**Confidence:** 3

**Review:**

This paper propose an extension of the Neural Theorem Provers (NTP) system that addresses the main issues of this method. The contributions of this paper allow to use this model on real-word datasets by reducing the time and space complexity of the NTP model.

Pro:

The paper is clear and well written and the contribution is relevant to ICLR. NTP systems by combining the advantages of neural models and symbolic reasoning are a promising research direction. Even though the results presented are lower than previous studies, they present the advantage of being interpretable.

Cons:

I'm not convinced by the model used to integrate textual mentions. The evaluation proposed in section 6.3 proposes to replace training triples by textual mention in order to evaluate the encoding module. However, it seems to me that, in this particular case, these mentions are very short sentences.  This could explained why such a simplistic model that simply average word embeddings is sufficient. I wonder if this would still work for more realistic (and thus longer) sentences.

Minor issues:

-Page 1: In particular [...] (NLU) and [...] (MR) in particular, ...

---

> ### Author Response · Authors · 2018-11-21
> **Reading module and textual mentions**
>
> Thank you very much for taking time to help bring our paper to a higher standard with your constructive feedback.
>
> > I'm not convinced by the model used to integrate textual mentions. These mentions are very short sentences.  This could explained why such a simplistic model that simply average word embeddings is sufficient.
>
> Thank you for pointing this out . We used a very simple reading model for showing that, even with an extremely simple approach, it is possible to integrate textual mentions while effectively improving results. This was a proof-of-concept demonstration on how a scalable end-to-end differentiable reasoning model enables reasoning over text while providing interpretable explanations for any given prediction (Sect. 6.3 and 6.4). It is true that, for Countries, textual mentions tend to be short, but it’s not the case for FB15k-237.
>
> Another motivation for using a simpler model is that it can perform on par or better than more complex model, thanks to a lower tendency to overfit to training data [1, 2] - we will emphasize this in the paper. We leave exploring more elaborate reading models to future work.
>
> [1] Arora et al, 2016, A simple but tough-to-beat baseline for sentence embeddings
> [2] White et al, 2015, How Well Sentence Embeddings Capture Meaning

---

### Official Review · AnonReviewer3 · 2018-11-03
**Interesting results, slightly unbalanced presentation**

**Rating:** 5
**Confidence:** 3

**Review:**

The authors propose several techniques to speed up the previously proposed Neural Theorem Prover approach. The techniques are evaluated via empirical results on several benchmark datasets.

Learning interpretable models is an important topic and the results here are interesting and valuable to the community. However, I feel that the paper in its current form is not yet ready for publication in ICLR, for the following reasons:

1) The authors propose three improvements. The first is a speed-up through nearest neighbor search instead of a brute-force search. This is the most elaborated section out of the three, yet seems like the most trivial -- unless the authors can provide an analytical bound on the loss in ntp score w.r.t the neighborhood size. It is a standard and well-known technique to restrict the search to a neighborhood, widely used in any applications of word embedding (e.g. in Khot et el's Markov Logic Networks for Natural Language Question Answering). The attention mechanism (essentially reducing the model capacity) is also well-known but its effect in this particular framework is not properly elaborated. The same can be said for the use of mentions.

2) The section on experiment results seems a bit rushed -- the authors did mention some last-minute discovery that may affect some of the presented results. The section can be a little hard to parse. In particular, it would be useful for the authors to focus on providing more insights on how the proposed techniques improve the results, and in what ways.

3) Section 2 on the NTP framework is not very helpful for a reader that has not read the previous paper on NTP (in particular, the part on training and rule learning). For a reader that has done so, the section feels redundant.

---

> ### Author Response · Authors · 2018-11-21
> **Improvements and related work**
>
> Many thanks for your constructive criticism - we greatly appreciate your efforts.
>
> > The first is a speed-up through nearest neighbor search instead of a brute-force search. This is the most elaborated section out of the three, yet seems like the most trivial
>
> The main focus of this work is making inference and learning in NTPs tractable. Previous to this work, training NTPs on large datasets simply unfeasible. Although it seems conceptually simple, we respectfully disagree that it is trivial to make NTPs’ end-to-end differentiable proving mechanism efficient by dynamically exploring only the most promising part of the proof space by means of dynamically pruning the computation graph at construction time, while still retaining computational efficiency superior to the original model. Furthermore, we extensively tested our improvements, and supported them with a large experimental assay.
>
> Importantly, this change enabled us to drastically increase the speed (more than two orders of magnitude in the case of Kinship and UMLS, and many more for larger datasets) while significantly decreasing the memory footprint of the model. Consequently, this enabled the application of explainable NTPs on large-scale text-enriched data - something that was simply not possible beforehand.
>
> This work is fundamentally different from Khot et al.’s paper. Although they use contextual similarities as a pre-processing step for defining the structure of a Markov Logic Network, they do not make use of embeddings. In contrast, we use ANNS on the embedding representations of facts and rules for identifying the most promising proof paths during the dynamic computation graph construction (forward pass). As for the word embedding search restriction to neighbourhoods, that is never utilised for for building the computation graph, but for the post-hoc analysis.
>
> The most related paper is [1], where Rae et al. use ANNS for computing a sparse attention distribution over memory entries. Their approach retains the representational power of the original memory networks, whilst training efficiently with very large memories. Similarly, our approach retains the expressiveness and the end-to-end differentiability of NTPs, while scaling to Knowledge Bases with millions of facts.
>
> [1] Rae et al., Scaling Memory-Augmented Neural Networks with Sparse Reads and Writes, 2016
>
> > [..] unless the authors can provide an analytical bound on the loss in ntp score w.r.t the neighborhood size.
>
> It is not trivial to provide an analytical bound on NaNTP: its derivation would depend on the characterisation of the approximation introduced by ANNS (still an open problem), and the greedy proof path selection, which may not yield to a globally optimal solution.
>
> NTPs follow a two-step process: i) given a query, they enumerate all possible proof paths, and ii) they compute a proof score for each proof path, returning the maximum proof score.
>
> After the proof path associated with the highest score is identified, the final score --- and ts gradient wrt. the model parameters --- can be computed exactly, since \nabla_\theta max(\rho_1, \ldots, \rho_n) = \nabla_\theta \rho_i, where \rho_i = max(\rho_1, \ldots, \rho_n). We clarify this in the paper.
>
> Exploring all possible proof paths is infeasible for large datasets, hence we propose using ANNS for greedily expanding only the most promising proof paths. This is motivated by observing that the proof score is given by the similarity between a goal and a fact or rule head: the higher the similarity, the higher the proof score.
>
> We can also see this problem in relation to the exploration vs exploitation trade-off: in Reinforcement Learning and optimisation it is fairly common to limit exploration to the most promising areas in the search space - instead of uniformly searching in the whole search space - at the risk of missing out high-reward regions. We analyse the cost of such a trade-off in our experiments, finding that our results are on par - or sometimes better than - the original model.
>
> Furthermore, we added an analysis on the impact of using ANNS in comparison with exact NNS and random neighbour selection, finding that ANNS is directly comparable with exact NNS but significantly faster. We added this characterisation to Table 8 in the Appendix.

---

> > ### Comment · AnonReviewer3 · 2018-12-04
> > **Some final comments**
> >
> > I appreciate the authors' effort in revising the paper and including additional experimental results. However, I feel that some of my concerns remain. There are obvious questions that beg for insights yet are simply ignored by the authors. For example, the computational speedup due to restricting the search space is obvious, so the numerical validation that it indeed results in significant speedup is not surprising. What seems surprising is that there is no price to pay in terms of the various performance metrics -- it even seems to mostly improve the performance. Why?
> >
> > I encourage the authors to prepare perhaps a journal version that fully explains NTP as well as the various improvements proposed here. This removes some of the space restrictions in a conference submission and allows a more detailed and hopefully insightful account of the approach.

---

> ### Author Response · Authors · 2018-11-21
> **Ablations, experiments, and background**
>
> > The attention mechanism (essentially reducing the model capacity) is also well-known but its effect in this particular framework is not properly elaborated. The same can be said for the use of mentions.
>
> Thank you for suggesting a more in-depth ablation study. We followed your advice and in order to assess the effect of attention to this framework, we added an ablation study on benchmark datasets, in Tables 6 and 7 of the appendix.
>
> Table 6 shows NaNTP with attention yielding higher average ranking accuracy and lower variances on Countries S1-3 and Kinship, with comparable performance on Nations and UMLS.
>
> In Table 7, we report the ablation results on two larger datasets - WN18 (141k facts) and WN18RR (87k facts). In the case of WordNet, using Attention for learning rules greatly increase the ranking accuracy - for instance, it increases from 83% hits@10 to 94% in the case of WN18, and from 25% to 43% in the case of WN18RR.
>
> In addition, Figure 2 combines the ablation study for both the effects of attention and added textual mentions. NaNTP with attention yields higher ranking accuracies with lower variances for Countries S1-3. As for the mentions, encoding them consistently improves the ranking accuracy in comparison to simply adding them as additional relations.
>
> > The authors did mention some last-minute discovery that may affect some of the presented results.
>
> The evaluation issue with the original NTP we discovered has now been fixed and we re-evaluated the original NTP models presented in the paper with the fixed evaluation.
>
> Consequently, we updated Table 3 with these scores and outlined why the scores differ to scores in the original paper. The results now fully testify that NaNTP is consistently better or, in the case of UMLS, on par with NTP.
>
> We reiterated over the experimental section, to improve clarity of exposition and focus on insights found.
>
> > Section 2 on the NTP framework is not very helpful for a reader that has not read the previous paper on NTP. For a reader that has done so, the section feels redundant.
>
> Thank you for pointing this out - we tried to hit a sweet spot between being self-contained and avoiding to replicate the NIPS 2017 paper about NTPs, but it was not an easy task. We rephrased Section 2 and shorten the redundant subsections, as you suggest.

---

### Official Review · AnonReviewer4 · 2018-11-11
**Interesting direction but needs more discussion**

**Rating:** 4
**Confidence:** 3

**Review:**

[Summary]
This paper scales NTPs by using approximate nearest neighbour search over facts and rules during unification. Additionally, the paper incorporates mentions as additional facts where the predicate is the text that the entities of the mention are contained in. The paper also suggests parameterizing predicates using attention over known predicates. The increments presented are reasonable and justified, but the experimental results, specifically on the larger datasets, warrant further investigation.

[Pros]
- Reasonable and interesting increments on top of NTP.
- Scaling the approach to larger datasets is well motivated.
- Utilizing text is an interesting direction for NTP in terms of integrating it with past work on KG completion.

[Cons]
- Empirical performance on larger datasets needs further investigation.
- No ablation study is performed so the effect of incorporating mentions and attention are unclear.
- Baseline performance on FB15k-237 seems weak compared to the original papers as well as more recent papers re-examining baselines for KG completion (http://aclweb.org/anthology/W17-2609). Is this due to the d=100 restriction, or were pretrained embeddings not used? Without further explanation, the claim that scores are competitive with SOTA seems unjustified, at least for FB15k-237 since the model performs significantly worse than the baselines which seem to be worse than previously reported.

[Comments]
- For reproducibility: it is unclear whether evaluation in FB15k-237 is carried out on the KB+Text, KB, or Text portions of the dataset.

[Overall]
It’s great that NTP was scaled up to handle larger datasets, however further analysis is needed. The argument that performance is given up for interpretability needs more discussion, and the effect of each addition to the system should be discussed as well.

---

> ### Author Response · Authors · 2018-11-21
> **Ablations and aims**
>
> Thank you for your constructive feedback.
> It is great to hear that you find this line of work interesting.
>
> > Empirical performance on larger datasets needs further investigation.
>
> First and foremost, we would like to highlight that the main focus of this paper is not climbing the link prediction leaderboards, but rather pushing NTP (a promising but until now computationally infeasible model) into practice by scaling it to large datasets, yielding results comparable with standard Neural Link Predictors, a class of models that was studied for nearly a decade now [1, 2]. Unlike Neural Link Predictors, NaNTPs can learn interpretable rules, as well as provide explanations for a given prediction, as we demonstrate in the experimental section. Moreover, they allow incorporating domain knowledge in the form of logic rules.
>
> > No ablation study is performed so the effect of incorporating mentions and attention are unclear.
>
> Following your advice on in-depth evaluation, we ran additional ablation studies for both the benchmark datasets and the large datasets.
>
> Table 6 in the Appendix shows that using attention in NaNTP for learning rule representations yields higher average ranking accuracy and lower variance on Countries S1-3 and Kinship, while yielding comparable results on Nations and UMLS.
>
> In Table 7, we report the ablation results on two larger datasets - WN18 (141k facts) and WN18RR (87k facts). In the case of WordNet, using attention for learning rules greatly increases the ranking accuracy. For instance, hits@10 increases from 83% to 94% in the case of WN18, and from 25% to 43% in the case of WN18RR.
>
> We hypothesise that this is because the attention has a constraining effect, regularising representations of the rules inside the convex hull of the representations of predicates.
>
> Furthermore, Figure 2 shows the ablation study of both the effect of attention and the added textual mentions. Consistently with Table 6, NaNTP with attention yields higher ranking accuracy and lower variance for Countries S1-3. As for the effect of reasoning over text, using distinct encoders for predicates and mentions consistently improves the ranking accuracy in comparison of simply using mentions as additional relation types.
>
> > Baseline performance on FB15k-237 seems weak compared to the original papers
>
> The lower baseline performance difference in neural link prediction baselines is mainly due to limiting the embedding size to 100 (d=100), and the number of training epochs to 100. These hyperparameters were used in the original NTP paper [3] and, for the sake of comparison to the original model, we decided to keep them fixed to the same values.
>
> Furthermore, exploring different embedding sizes for NaNTP was prohibitive due to a lack of computation resources. In NaNTPs after the ANNS index construction, the complexity of inference (and thus learning) grows logarithmically in the size of the Knowledge Base, and evaluating the ranking of each single test triple requires scoring all its possible corruptions (i.e. 82k triples on WN18): this is a very computationally expensive procedure even for neural link predictors.
>
> In the case of FB15k-237, the experiment also involved the textual mentions proposed in [3]. We corrected this in the revised version of the paper.
>
> [1] Paccanaro et al., Learning Distributed Representations of Concepts using Linear Relational Embedding, IEEE Transactions on Knowledge and Data Engineering 2000
> [2] Bordes et al., Translating Embeddings for Modeling Multi-relational Data, NIPS 2013
> [3] Rocktaschel et al., End-to-end Differentiable Proving, NIPS 2017
> [4] Toutanova et al., Representing text for joint embedding of text and knowledge bases, EMNLP 2015

---

> > ### Comment · AnonReviewer4 · 2018-11-21
> > **Thanks for the edits! Remaining concern below**
> >
> > Thanks for taking the time to make edits. Although the ablation studies are indeed an improvement and address half of my concerns, they are not enough for me to change my score.
> >
> > I’d like to reiterate that I believe this direction is important and interesting, but my remaining concern is the following: The question the paper answers is “can the NaNTP be run on large datasets?” However, the question I would like answered is not only that, but also “if I were to tackle multi-hop link prediction at scale, should I use the NaNTP over other uninterpretable methods?” The other methods should include not only neural link prediction, but also other multi-hop methods.
> >
> > I believe this would be a more solid contribution if the paper:
> >
> > 1) demonstrated competitive-to-SOTA performance as reported in other papers (not just the re-implementation of the baselines), as one worry is that the baselines scores are not strong enough. Kadlec et. al [1] note that models trained on FB15K (although different than FB15K-237, the same statement likely applies) are very sensitive to hyperparameters. This would entail either tuning the baselines to match previous numbers or simply using previously reported numbers if computational resources are not available, then improving the performance of the NaNTP if possible. If not possible, provide a convincing explanation of the results. An example of this is in Das et. al. [3], where they justified the superior performance of embedding methods vs path-based methods on FB15K-237 in section 3.1.2.
> >
> > 2) includes comparisons to more recent work on multi-hop link prediction such as Minerva [3], Diva [2], etc., where the comparison includes ideally both speed and KB metrics.
> >
> > I am convinced that NaNTP can be scaled; however, I would like a clearer picture of how it compares to related models.
> >
> > [1] Kadlec, Bajgar, Kleindienst. Knowledge Base Completion:Baselines Strike Back. https://aclanthology.info/papers/W17-2609/w17-2609.
> > [2] Chen, Xiong, Fan, Wang. Variational Knowledge Graph Reasoning. https://aclanthology.coli.uni-saarland.de/papers/N18-1165/n18-1165
> > [3] Das et. al. Go for a Walk and Arrive at the Answer: Reasoning Over Paths in a Knowledge Bases using Reinforcement Learning. https://openreview.net/forum?id=Syg-YfWCW

---

> > > ### Author Response · Authors · 2018-11-26
> > > **Baselines, ablations, and experiments**
> > >
> > > Thank you for your answer,
> > >
> > > > “if I were to tackle multi-hop link prediction at scale, should I use the NaNTP over other uninterpretable methods?” --- I am convinced that NaNTP can be scaled; however, I would like a clearer picture of how it compares to related models.
> > >
> > > You are completely right, thanks for pointing this out. In Table 1 and Table 3 we added several baselines from the literature (DistMult, ComplEx, ConvE, NeuralLP, and MINERVA, from Das et al.’s “Go for a Walk and Arrive at the Answer” paper), and ablations on attentions and text. We added an official comment enumerating our changes to the revised version of the paper.
> > >
> > > > demonstrated competitive-to-SOTA performance as reported in other papers
> > >
> > > In Table 3 we show that NaNTP is competitive, and often better, than the original NTP [1] on Countries (S1-S3), Kinship, Nations, and UMLS, while being several orders of magnitude faster on the reference datasets. NTP was yielding better results than SOTA methods such as ComplEx,  while being able to provide explanations for its predictions. Results for NTP on WN18, WN18RR, and Freebase are not available, since NTP does not scale to such Knowledge Bases.
> > >
> > > In the revised paper, we report comparisons with DistMult, ComplEx, ConvE, NeuralLP, and MINERVA, showing that NaNTP yields comparable results. Please note that Neural Link Predictors such as DistMult and ComplEx belong to a family of Representation Learning models that was studied for a decade now [2, 3], while Neural Theorem Provers started gaining momentum in recent months, one main limitation being their scalability. Since Neural Theorem Provers were a less explored area of research, we think they are more likely to have less polished results than better explored areas, such as Neural Link Predictors.
> > >
> > > [1] Rocktäschel and Riedel. End-to-End Differentiable Proving. NIPS 2017
> > > [2] Paccanaro and Hinton. Learning Distributed Representations of Concepts Using Linear Relational Embedding. TKDE 2001
> > > [3] Bordes et al. Translating Embeddings for Modeling Multi-relational Data. NIPS 2013

---

> > > > ### Comment · AnonReviewer4 · 2018-11-26
> > > > **Clarification on previous concern: discussion is more than just numbers**
> > > >
> > > > Thanks for the update. I wanted to clarify that the numbers themselves are not the reason for my previously listed concern and score, but the lack of analysis beyond just the numbers themselves on the large-scale datasets.
> > > >
> > > > I gave Das et. al.'s section on FB15k-237 as an example of giving discussion beyond just the numbers. I would like to see a stronger *analysis* of the results on the larger datasets, and an explanation for the numbers. The numbers themselves not reaching SOTA is fine, and does not affect the score I give. To be more concrete, I would like to see
> > > >
> > > > 1) some representative examples where single-link prediction does well and NaNTP fails (with an analysis of why NaNTP does not do as well and evidence of the example's representativeness), and ideally a conjecture about possible future work to bridge the gap.
> > > >
> > > > 2) Also include some examples where NaNTP does well but single-link does not (in addition to an analysis of why).
> > > >
> > > > I'm looking for a statement like "NaNTPs overall perform worse, but do much better on this class of examples but much worse on this class," with supporting examples and justification for claims. So no re-training is necessary; I would mainly like to see a deeper comparison at evaluation time. The reason for this is that, in the worst case, NaNTP simply does worse on all classes of examples regardless of how the classes are chosen in the large-data setting. This would definitely be worth reporting (and my review will not penalize the paper's score for reporting negative results). Are we seeing something similar to Naive Bayes vs Logistic regression, where NB does better in the small-data regime but not as well in the large data regime? Hopefully that is not the case, and your analysis will lead to future work on how to bridge the performance gap. The original NTP paper already argued that its interpretable nature was interesting and was able to learn transitivity, hopefully there is more analysis to be done than just restating their claims.
> > > >
> > > > I think the term polish in the ICLR guideline is referring to the numbers, hopefully the expectations for analysis of results are still equally high for all papers regardless of topic. Again, I am fine with the numbers, but expect more analysis *beyond* the numbers. I am excited to see the analysis, good luck!

---

> > > > > ### Author Response · Authors · 2018-11-27
> > > > > **Analysis and discussion, part 2**
> > > > >
> > > > > The benefit of a clear logical structure is even more evident in the case of WN18, which is characterised by a more logical relational structure. For instance, by learning clear rules such as part_of(X, Y) :- has_part(Y, X), hyponym(X, Y) :- hypernym(Y, X), and hypernym(X, Y) :- hyponym(Y, X), NaNTP can accurately predict the underlying structure in WN18, and use this knowledge to yield more accurate link prediction results than ComplEx in several cases, as present in the table above.
> > > > >
> > > > > However, the opposite is also true: we can see that, in some cases, logic rules and continuous unification may are not sufficient for some of the link prediction tasks. For instance, on WN18, NaNTP was not able to learn a set of rules for accurately predicting the _derivationally_related_form predicate and, for such a reason, ComplEx can yield a higher accuracy on this type of relations.
> > > > >
> > > > > On the other hand, predictions ComplEx yields are not as easy to explain, since the score is a function of the embeddings of the entities involved in the prediction. On WN18RR, ComplEx shines on relations that reflect the cluster structure of the network, such as _also_see and _derivationally_related_form: as it does not need to rely on an underlying logical structure (as NaNTP), it can more accurately handle the cases where such a structure is missing.
> > > > >
> > > > > However, ComplEx yields less accurate results on relations which can be accurately predicted by leveraging an underlying logical structure, which NaNTP can learn and then leverate at test time. For instance, on WN18, ComplEx is less accurate than NaNTP on predicates such as _hypernym (logically related to _hyponym), _part_of (related to _has_part), _hyponym (related to _hypernym) and _member_holonym (related to _member_meronym).
> > > > >
> > > > > Given that ComplEx and NaNTP have complementary strengths (and weaknesses), we believe the gap between them can be narrowed down by using ComplEx or any other link prediction algorithm as a regulariser (akin to the NTP-lambda in the original NTP paper), by proposing a mixture of experts, and possibly by adding a mixture of correctly induced rules from multiple runs of NaNTP.

---

> > > > > ### Author Response · Authors · 2018-11-27
> > > > > **Analysis and discussion, part 1**
> > > > >
> > > > > We see your point and wholly agree with it. Sadly, since we cannot update the paper anymore, we provide our analysis and comparisons here.
> > > > >
> > > > > We started the analysis with the per-predicate comparison of NaNTP and ComplEx (our best performing models for both), in terms of Mean Reciprocal Rank (MRR), on both WN18 and WN18RR.
> > > > >
> > > > > In the following, we provide the per-predicate MRR results on WN18 (* denotes cases where NaNTP performs better or on par as ComplEx):
> > > > >
> > > > > WN18							NaNTP	ComplEx
> > > > > _hyponym						*0.937	0.890
> > > > > _member_holonym				*0.912	0.809
> > > > > _hypernym						*0.934	0.891
> > > > > _part_of							*0.921	0.826
> > > > > _derivationally_related_form		0.035	0.917
> > > > > _member_of_domain_topic		0.722	0.745
> > > > > _instance_hyponym				0.490	0.776
> > > > > _synset_domain_topic_of			*0.771	0.746
> > > > > _synset_domain_region_of			0.362	0.689
> > > > > _member_of_domain_region		0.417	0.667
> > > > > _has_part						0.680	0.839
> > > > > _also_see						*0.554	0.511
> > > > > _instance_hypernym				0.645	0.774
> > > > > _member_meronym				0.614	0.815
> > > > > _verb_group						*0.951	0.677
> > > > > _synset_domain_usage_of			0.775	0.776
> > > > > _member_of_domain_usage		*0.769	0.722
> > > > > _similar_to						*1.000	*1.000
> > > > >
> > > > > Next, we provide the per-predicate MRR results on WN18RR:
> > > > >
> > > > > WN18RR					NaNTP	ComplEx
> > > > > _hypernym					0.022	0.092
> > > > > _derivationally_related_form	0.934	0.941
> > > > > _member_meronym			0.055	0.133
> > > > > _has_part					0.046	0.123
> > > > > _also_see					*0.593	0.522
> > > > > _member_of_domain_region	0.011	0.040
> > > > > _verb_group					*0.893	0.825
> > > > > _synset_domain_topic_of		0.042	0.184
> > > > > _instance_hypernym			0.093	0.241
> > > > > _member_of_domain_usage	0.030	0.201
> > > > > _similar_to					0.764	1.000
> > > > >
> > > > > From the results, we can see that NaNTP and ComplEx have complementary strengths and weaknesses. For instance, by inspecting the rules learned by NaNTP on WN18RR, we can see NaNTP learns symmetry rules such as:
> > > > >
> > > > > _derivationally_related_form(X0, Y0) :- _derivationally_related_form(Y0, X0)
> > > > > _similar_to(X0, Y0) :- _similar_to(Y0, X0)
> > > > >
> > > > > The _derivationally_related_form rule is often used by our model and, as a result of that, it makes NaNTP as accurate as ComplEx on the _derivationally_related_form predicate. The _similar_to rule is interesting as, though it does hold in general, the NaNTP model never uses it when predicting with _similar_to relations. This is a direct consequence of the way WN18RR was created, as these particular examples are filtered out of the dev and test sets. The same rule is induced in WN18 where it is fully utilised. Instead of this rule, NaNTP on WN18RR is using another learned rule:
> > > > >
> > > > > _verb_group(X0, Y0) :- _also_see(Y0, X0).
> > > > >
> > > > > This is quite interesting, as the same rule, is often used to express multiple symmetrical relationship with specific predicates such as _also_see, _verb_group and _similar_to.
> > > > >
> > > > > This shows that the result of the originally proposed decoding of the rule with a one-nearest-neighbor (1-NN), though informative, should not be taken as a literal discrete rule. However, although we do not have a concrete representation of a rule as we might wish, we can still decide whether that rule is meaningful or not, and use such insights for refining the model, improving our understanding of the domain, or providing explanations for any given prediction. Moreover, when decoding rules, looking at the final proof paths is highly informative. For example, the following (correct) proof paths:
> > > > >
> > > > > _also_see(coherent.a.01, logical.a.01) is explained by _verb_group(X0, Y0) :- _also_see(Y0, X0) and _also_see(logical.a.01, coherent.a.01)
> > > > >
> > > > > _verb_group(allow.v.03, permit.v.01) is explained by _verb_group(X0, Y0) :- _also_see(Y0, X0) and _verb_group(permit.v.01, allow.v.03)
> > > > >
> > > > > _similar_to(dynamic.a.01, hold-down.n.01) is explained by _verb_group(X0, Y0) :- _also_see(Y0, X0) and _similar_to(hold-down.n.01, dynamic.a.01)
> > > > >
> > > > > essentially tell us that the rule at question is used for representing symmetry.
> > > > >
> > > > > All in all, NaNTP can learn symmetry rules, while also softly unifying related predicates and by leveraging such rules can perform better or on par with ComplEx on relations exhibiting a clear logical structure (symmetric relations), while still benefiting from the continuous unification.

---

> ### Author Response · Authors · 2018-11-30
> **Some final comments**
>
> Dear Reviewer 4,
>
> Thank you for being so active in participating in the discussion and rebuttal period with us. Through our exchange, we've been able to make significant improvements to the paper, provide additional results, and expand our analysis of our method in comments which will be added to the final revision of the paper:
> * https://openreview.net/forum?id=BJzmzn0ctX&noteId=HkxqI2-oAm
> * https://openreview.net/forum?id=BJzmzn0ctX&noteId=B1eBEhWs0m
>
> We aim to have addressed your concerns and believe we have made the paper significantly stronger in response to your advice. We hope now that you will consider adjusting your score to reflect you reevaluation of our paper in light of the improvements made. At very least we hope you will provide us with constructive criticism as to where further improvements can be made if you feel the paper still falls short despite additional experiments and analysis, but naturally we hope you will agree the paper is now strong enough to be accepted.

---

### Author Response · Authors · 2018-11-26
**Baselines, ablations, improvements in clarity, and experiments.**

We thank all reviewers for insightful and very detailed feedback. We followed their suggestions, and updated our submission as follows:
- We added a series of comparisons with state-of-the-art link prediction methods (namely DistMult, ComplEx, ConvE, NeuralLP, and MINERVA) in Table 1 and Table 3.
- In Table 1 we also added a series of ablations, for analysing the impact of using attention (on WN18, WN18RR, and FB15k-237.E) and natural language surface forms (on FB15k-237.E), and analysed them in Section 6.
- We greatly improved clarity in the section introducing Neural Theorem Proving.
- We added more results in the Appendix - Table 6, Table 7, and Table 8 - including additional ablations on using attention and text, showing that both helps improving the model’s predictive accuracy.
- We also added a comparison between Exact and Approximate Nearest Neighbour Search (ANNS), and Random Neighbourhood (Table 8), showing that ANNS yields results on par with Exact NNS, while being orders of magnitude more computationally efficient, in terms of both time and space complexity.
- We discussed the issues we found in the evaluation function provided with the code of the NIPS 2017 paper introducing NTPs, and re-computed their experiments.

---

### Meta-Review · Area_Chair1 · 2018-12-17
**Worthy goal, but limited novelty and analysis**

**Confidence:** 3
**Recommendation:** Reject

**Metareview:**

This paper focuses on scaling up neural theorem provers, a link prediction system that combines backward chaining with neural embedding of facts, but does not scale to most real-world knowledge bases. The authors introduce a nearest-neighbor search-based method to reduce the time/space complexity, along with an attention mechanism that improves the training. With these extensions, they scale NTP to modern benchmarks for the task, including ones that combine text and knowledge bases, thus providing explanations for such models.

The reviewers and the AC note the following as the primary concerns of the paper: (1) the novelty of the contributions is somewhat limited, as nearest neighbor search and attention are both well-known strategies, as is embedding text+facts jointly, (2) there are several issues in the evaluation, in particular around analysis of benefits of the proposed work on new datasets. There were a number of other potential weaknesses, such the performance on some benchmarks (Fb15k) and clarity and writing quality of a few sections.

The authors provided significant revisions to the paper that addressed many of the clarity and evaluation concerns, along with providing sufficient comments to better contextualize some of the concerns. However, the concerns with novelty and analysis of the results still hold. Reviewer 3 mentions that it is still unclear in the discussion why the accuracy of the proposed approach matches/outperforms that of NTP, i.e. why is there not a tradeoff. Reviewer 4 also finds the analysis lacking, and feels that the differences between the proposed work and the single-link approaches, in terms of where each excels, are described in insufficient detail. Reviewer 4 focused more on the simplicity of the text encoding, which restricts the novelty as more sophisticated text embeddings approaches are commonplace.

Overall, the reviewers raised different concerns, and although all of them appreciated the need for this work and the revisions provided by the authors, ultimately feel that the paper did not quite meet the bar.